# MULTIPLE-PLAY STOCHASTIC BANDITS WITH PRIORITIZED ARM CAPACITY SHARING

## ABSTRACT

This paper proposes a variant of multiple-play stochastic bandits tailored to resource allocation problems arising from LLM applications, edge intelligence applications, etc. The proposed model is composed of $M$ arms and $K$ plays. Each arm has a stochastic number of capacities, and each unit of capacity is associated with a reward function. Each play is associated with a priority weight. When multiple plays compete for the arm capacity, the arm capacity is allocated in a larger priority weight first manner. Instance independent and instance dependent regret lower bounds of $\Omega(\alpha_1 \sigma \sqrt{KMT})$ and $\Omega(\alpha_1 \sigma^2 \frac{MK}{\Delta} \ln T)$ are proved, where $\alpha_1$ is the largest priority weight and $\sigma$ characterizes the reward tail. When model parameters are given, we design an algorithm named `MSB-PRS-OffOpt` to locate the optimal play allocation policy with a computational complexity of $O(M^3 K^3)$. Utilizing `MSB-PRS-OffOpt` as a subroutine, an approximate upper confidence bound (UCB) based algorithm is designed, which has instance independent and instance dependent regret upper bounds matching the corresponding lower bound up to factors of $K\sqrt{\ln KT}$ and $\alpha_1 K$ respectively. To this end, we address nontrivial technical challenges arising from optimizing and learning under a special nonlinear combinatorial utility function induced by the prioritized resource sharing mechanism.

## 1 INTRODUCTION

The Multi-play multi-armed bandit (MP-MAB) is a classical sequential learning framework Anantharam et al. (1987a). The canonical MP-MAB model consists of one decision maker who pulls multiple arms per decision round. Each pulled arm generates a reward, which is drawn from an unknown probability distribution. The objective is to maximize the cumulative reward facing the exploration vs. exploitation dilemma. MP-MAB frameworks are applied to various applications such as online advertising Lagrée et al. (2016); Komiyama et al. (2017); Yuan et al. (2023), power system Lesage-Landry & Taylor (2017), mobile edge computing Chen & Xie (2022); Wang et al. (2022a); Xu et al. (2023), etc. Recently, various variants of MP-MAB were studied, which tap potentials of MP-MAB framework for resource allocation problems and advance the bandit learning literature Chen & Xie (2022); Moulos (2020); Xu et al. (2023); Wang et al. (2022a); Yuan et al. (2023).

This paper extends MP-MAB to capture the prioritized resourcing sharing mechanism, contributing a fine-grained resource allocation model. We aim to reveal fundamental insights on the interplay of this mechanism and learning. Prioritized resourcing sharing mechanisms are implemented in a large class of resource allocation problems arising from mobile edge computing Chen et al. (2021); Gao et al. (2022); Ouyang et al. (2019; 2023), ride sharing Chen & Xie (2022), etc., and have the potential to enable differentiated services in LLM applications. For example, in LLM applications, reasoning tasks and LLM instances can be modeled as plays and arms respectively. Multiple LLM reasoning tasks (plays) share an instance of LLM (an arm) according to their priority quantified by price, membership hierarchy, etc. In mobile edge computing systems, the infrastructure of edge intelligence, tasks and edge servers can be modeled as plays and arms respectively. When multiple tasks (or plays) are offloaded to the same edge server (or arm), the available computing resource is shared among them according to the differentiated pricing mechanism (an instance of prioritized resourcing sharing mechanism).

Formally, we proposed MSB-PRS (Multiple-play Stochastic Bandits with Prioritized Resource Sharing). The MSB-PRS is composed of $K \in \mathbb{N}_+$ plays and $M \in \mathbb{N}_+$ arms. Each play has a priority weight and movement costs. Each arm has a stochastic number of units of capacities. Plays share the capacity in a high priority weight first manner. A play receives a reward scaled by its weight only when it occupies one unit of capacity. The objective is to maximize the cumulative utility (rewards minus costs) in $T \in \mathbb{N}_+$ rounds. Some recent works tailored MP-MAB to the same or similar applications Chen & Xie (2022); Xu et al. (2023); Wang et al. (2022a;b); Yuan et al. (2023). The key difference to this line of research is on the stochastic capacity with bandit feedback and prioritized capacity sharing. This difference poses new challenges. One challenge lies in locating the optimal play allocation policy. The movement cost and the prioritized capacity sharing impose a nonlinear combinatorial structure on the utility function, which hinders locating the optimal allocation. In contrast to previous works Xu et al. (2023); Wang et al. (2022a;b), top arms do not warrant optimal allocation. This nonlinear combinatorial structure also makes it difficult to distinguish optimal allocation from sub-optimal allocation from feedback. As a result, it is nontrivial to balance the exploring vs. exploitation tradeoff. We address these challenges.

## 1.1 CONTRIBUTIONS

**Model and fundamental learning limits.** We formulate MSB-PRS, which captures the prioritized resourcing sharing nature of resource allocation problems. We prove instance independent and instance dependent regret lower bounds of $\Omega(\alpha_1 \sigma \sqrt{KMT})$ and $\Omega(\alpha_1 \sigma^2 \frac{M}{\Delta} \ln T)$ respectively. Technically, we tackle the aforementioned nonlinear combinatorial structure challenge by identifying one special instances of the MSB-PRS that are composed of carefully designed multiple independent groups of classical multi-armed bandits and batched MP-MAB.

**Efficient learning algorithms. (1) Computational efficiency.** Given model parameters, to tackle the computational challenge of locating the optimal play allocation policy, we characterize the aforementioned nonlinear combinatorial structure by constructing a priority ranking aware bipartite graph. A connection between the utility of arm allocation policies and the saturated, monotone and priority compatible matchings is established. This connection enables us to design `MSB-PRS-OffOpt`, which locates the optimal play allocation policy with a complexity $O(M^3 K^3)$ from a search space with size $K^M$. **(2) Sample efficiency.** Utilizing `MSB-PRS-OffOpt` as a subroutine, we design an approximate UCB based algorithm, which reduces the per-round computational complexity of the exact UCB based algorithm from $K^M$ to $O(K^3 M^3)$. We prove sublinear instance independent and instance dependent regret upper bounds matching the corresponding lower bounds up to factors of $K\sqrt{\ln KT}$ and $\alpha_1 K^2$ respectively. The key proof idea is exploiting the monotone property of the utility function to: (1) prove the validity of the approximate UCB index; (2) show suboptimal allocations make progress in improving the estimation accuracy of poorly estimated parameters, which gear the learning algorithm toward identifying more favorable play allocation policies.

## 2 RELATED WORK

Anantharam *et al.* Anantharam et al. (1987a) proposed the canonical MP-MAB model, where they established an asymptotic lower bound on the regret and designed an algorithm achieving the lower bound asymptotically. Komiyama *et al.* Komiyama et al. (2015) showed that Thompson sampling achieves the regret lower bound in the finite time sense. Anantharam *et al.* Anantharam et al. (1987b) extended the canonical MP-MAB model from IID rewards to Markovian rewards. This Markovian MP-MAB model was further extended to the rested bandit setting Moulos (2020). MP-MAB with a reward function depending on the order of plays was studied in Lagrée et al. (2016); Komiyama et al. (2017). This reward function was motivated by clicking the model of web applications. They established lower bounds on the regret and designed a UCB based algorithm to balance the exploration vs. exploitation tradeoff. MP-MAB with switching cost is studied in Agrawal et al. (1990); Jun (2004). They proved the lower bound on the regret and designed algorithms that achieve the lower bound asymptotically. MP-MAB with budget constraint is considered in Luedtke et al. (2019); Xia et al. (2016); Zhou & Tomlin (2018) and a stochastic number of plays in each round is considered in Lesage-Landry & Taylor (2017), which is motivated by power system. Recently, Yuan *et al.* Yuan et al. (2023) extended the canonical MP-MAB classical to the sleeping bandit setting, for the purpose of being tailored to the recommender systems.

Our work is closely related to Chen & Xie (2022); Wang et al. (2022a); Xu et al. (2023). Chen *et al.* Chen & Xie (2022) tailored the canonical MP-MAB model for the user-centric selection problems. Their model considered homogeneous plays and expert feedback on capacity. They designed a Quasi-UCB algorithm for this problem with sublinear regret upper bounds. Our work generalizes their model to capture heterogeneous plays, prioritized resourcing sharing, and bandit feedback on the capacity. This extension not only be more friendly to real-world applications, but also incurs new challenges for locating the optimal allocation and design learning algorithms. We design a UCB based algorithm and prove both regret upper bounds and lower bounds. Wang *et al.* Wang et al. (2022a) proposed a model that also allowed multiple plays to share capacity on an arm. Their model considers a deterministic capacity provision. The capacity is unobservable and coupled with the reward. They proved regret lower bound on regret and designed an action elimination based algorithm whose regret matches the regret lower bound to a certain level. Xu *et al.* Xu et al. (2023) extended this model to the setting with strategic agents and competing for the capacity. They analyzed the Nash equilibrium in the offline setting and proposed a Selfish MP-MAB with an Averaging Allocation approach based on the equilibrium.

Various works share some connections to the MP-MAB research line. Combinatorial bandits Cesa-Bianchi & Lugosi (2012); Chen et al. (2013); Combes et al. (2015b) generalize the reward function of the canonical MP-MAB from linear to non-linear. Various variants of combinatorial bandits were studied: (1) combinatorial bandits with semi-bandit feedback Chen et al. (2013; 2016); Gai et al. (2012); Combes et al. (2015b), i.e., the reward of each pulled arm is revealed; (2) combinatorial bandits with bandit feedback: Cesa-Bianchi & Lugosi (2012); Combes et al. (2015b), i.e., only one reward associated with the pulled arm set is revealed; (3) combinatorial bandits with different combinatorial structures, i.e., matroid Kveton et al. (2014), $m$-set Anantharam et al. (1987a), permutation Gai et al. (2012), etc. Cascading bandit Combes et al. (2015a); Kveton et al. (2015b); Wen et al. (2017) extends the reward function of the canonical MP-MAB from linear to a factorization form over the set of selected arms. Decentralized MP-MAB (a.k.a. multi-player MAB) Agarwal et al. (2022); Anandkumar et al. (2011); Rosenski et al. (2016); Bistritz & Leshem (2018); Wang et al. (2020)) considers the setting that players either cannot communicate with others or their communication is restrictive.

# 3 MSB-PRS MODEL

## 3.1 MODEL SETTING

For any integer $N$, the notation $[N]$ denotes a set $[N] \triangleq \{1, \ldots, N\}$. The MSB-PRS consists of one decision maker, $M \in \mathbb{N}_+$ arms, $K \in \mathbb{N}_+$ plays and a finite number of $T \in \mathbb{N}_+$ decision rounds. In each decision round $t \in [T]$, the decision maker needs to assign all $K$ plays to arms. Each play can be assigned to one arm, and multiple plays can be allocated to the same arm. The objective is to maximize the total utility, whose formal definition is deferred after the arm model and reward model are made clear.

**Arm model.** The arm $m \in [M]$ is characterized by a pair of random variables $(D_m, R_m)$, where $D_m$ characterizes the stochastic availability of capacity and $R_m$ characterizes the per unit capacity rewards. The support of $D_m$ is a subset of $[d_{\max}]$, where $d_{\max} \in \mathbb{N}_+$ denotes the maximum possible units of capacity on an arm. Let $D_m^{(t)}$ denote the number of units of capacity available on arm $m$ in round $t$. The $D_m^{(t)}$ is drawn from $D_m$, i.e., $D_m^{(t)} \sim D_m$, and each $D_m^{(t)}$ drawn from $D_m$ is independent across $t$ and $m$. The $i$-th unit of capacity on arm $m$ is associated with a reward denoted by $R_{m,i}^{(t)}$, where $i \in [D_m^{(t)}]$. The $R_{m,i}^{(t)}$ is drawn from $R_m$, i.e., $R_{m,i}^{(t)} \sim R_m$ whose support is a subset of $\mathbb{R}$, and each $R_{m,i}^{(t)}$ drawn from $R_m$ is independent across $t, m$ and $i$. Denote the mean of $R_m$ as $\mu_m \triangleq \mathbb{E}[R_m]$. Without loss of generality, we assume $\mu_m > 0, \forall m \in [M]$. We assume that $R_m$ is $\sigma$-subgaussian, where $\sigma \in \mathbb{R}_+$. Let $\boldsymbol{\mu} \triangleq [\mu_m : \forall m \in [M]]$ denote the reward mean vector. Let $\boldsymbol{P}_m \triangleq [P_{m,d} : \forall d \in [d_{max}]]$ denote the complementary cumulative probability vector of $D_m$, where

$$P_{m,d} = \mathbb{P}[D_m^{(t)} \geq d], \forall d \in [d_{max}], m \in [M].$$

For presentation convenience, denote the complementary cumulative probability matrix as:

$$\boldsymbol{P} \triangleq [P_{m,d} : \forall d \in [d_{max}], m \in [M]].$$

The $\boldsymbol{\mu}$ and $\boldsymbol{P}$ are unknown to the decision maker. Arms can model instances of LLM, edge servers, etc (refer to Section 1).

**Play and priority model.** The play $k \in [K]$ is characterized by $(\boldsymbol{c}_k, \alpha_k)$, where $\boldsymbol{c}_k \in (\mathbb{R}_+ \cup \{+\infty\})^M$ and $\alpha_k \in \mathbb{R}_+$. The $\boldsymbol{c}_k$ denotes the movement cost vector associated with play $k$ and denote its entries as $\boldsymbol{c}_k \triangleq [c_{k,m} : \forall m \in [M]]$, where $c_{k,m} \in \mathbb{R}_+ \cup \{+\infty\}$ denotes the movement cost of assigning play $k$ to arm $m$. The case $c_{k,m} = +\infty$ models the constraint that arm $m$ is unavailable to play $k$. The weight $\alpha_k$ quantifies the priority of play $k$. Larger weight implies higher priority. Without loss of generality, we assume

$$\alpha_1 \geq \alpha_2 \geq \cdots \geq \alpha_K > 0.$$

The $\alpha_k$'s capture differentiated service of mobile edge computing, or the superiority of cars in ride sharing systems. Both $\boldsymbol{c}_k$ and $\alpha_k$ are known to the decision maker. Plays can model reasoning tasks, computing tasks, etc (refer to Section 1).

**Prioritized capacity sharing model.** Let $a_k^{(t)} \in [M]$ denote the arm pulled by play $k \in [K]$ in round $t$. Denote the play allocation or action profile in round $t$ as $\boldsymbol{a}^{(t)} \triangleq [a_k^{(t)} : \forall k \in [K]]$. Denote the number of plays assigned to arm $m$ in round $t$:

$$N_m^{(t)} \triangleq \sum_{k \in [K]} \mathbb{1}_{\{a_k^{(t)} = m\}}$$

Plays are prioritized according to their weights. Specifically, in round $t$, the $N_m^{(t)}$ plays assigned to arm $m$ are ranked according to their weights, i.e., $\alpha_k$'s, in descending order, where ties are broke arbitrarily, and they share the capacity according to this order. Consider a play assigned to arm $m$, i.e., $a_k^{(t)} = m$, denote its rank on arm $m$ as $\ell_k^{(t)} \in [K]$. In round $t$, only top-$\min\{N_m^{(t)}, D_m^{(t)}\}$ plays assigned to arm $m$ are allocated capacities, in a fashion that one unit of capacity per play. Namely, when the capacity is abundant, i.e., $D_m^{(t)} \geq N_m^{(t)}$, the $D_m^{(t)} - N_m^{(t)}$ units of capacity are left unassigned; and when the capacity is scarce, i.e., $D_m^{(t)} < N_m^{(t)}$, the $N_m^{(t)} - D_m^{(t)}$ plays do not get capacity.

**Rewards and feedback.** Once play $k$ gets a unit of capacity, a reward scaled by the weight is generated:

$$X_k^{(t)} \triangleq \begin{cases} \alpha_k R_{m, \ell_k^{(t)}}^{(t)}, & \text{if } \ell_k^{(t)} \leq D_{a_k^{(t)}}^{(t)}, \\ \text{null}, & \text{otherwise.} \end{cases}$$

where null models that play $k$ does not receive any reward when it does not occupy any capacity. The decision maker observes the rewards received by each arm. Let $\boldsymbol{X}^{(t)} \triangleq [X_k^{(t)} : \forall k \in [K]]$ denote the reward vector observed in round $t$. In round $t$, the number of capacity $D_m^{(t)}$ is revealed to the decision maker if and only if at least one play is assigned to arm $m$ in this round, i.e., $N_m^{(t)} > 0$. Denote the capacity feedback vector $\boldsymbol{D}^{(t)} \triangleq [D_m^{(t)} : m \in \{m' | N_{m'}^{(t)} > 0\}]$. The decision maker observes $\boldsymbol{X}^{(t)}$ and $\boldsymbol{D}^{(t)}$ in round $t$.

## 3.2 PROBLEM FORMULATION

Denote the expected total reward generated from arm $m$ in round $t$ as $\overline{R}_m(\boldsymbol{a}^{(t)}; \mu_m, \boldsymbol{P}_m)$, formally:

$$\overline{R}_m(\boldsymbol{a}^{(t)}; \mu_m, \boldsymbol{P}_m) \triangleq \mathbb{E}\left[ \sum_{k \in [K]} \mathbb{1}_{\{a_k^{(t)} = m\}} X_k^{(t)} \right] = \mu_m \sum_{k \in [K]} \mathbb{1}_{\{a_k^{(t)} = m\}} \alpha_k P_{m, \ell_k^{(t)}}.$$

Let $U_m(\boldsymbol{a}^{(t)}; \mu_m, \boldsymbol{P}_m)$ denote the expected utility earned from arm $m$ in round $t$. It is defined as the expected reward minus the movement cost, formally:

$$U_m(\boldsymbol{a}^{(t)}; \mu_m, \boldsymbol{P}_m) \triangleq \overline{R}_m(\boldsymbol{a}^{(t)}; \mu_m, \boldsymbol{P}_m) - \sum_{k \in [K]} c_{k,m} \mathbb{1}_{\{a_k^{(t)} = m\}}.$$

Let $U(\boldsymbol{a}^{(t)}; \boldsymbol{\mu}, \boldsymbol{P})$ denote the aggregate utility from all plays given action profile $\boldsymbol{a}_t$, formally:

$$U(\boldsymbol{a}^{(t)}; \boldsymbol{\mu}, \boldsymbol{P}) \triangleq \sum_{m \in [M]} U_m(\boldsymbol{a}^{(t)}; \mu_m, \boldsymbol{P}_m). \tag{1}$$

The objective is to maximize the total utility in $T$ rounds, i.e., maximize $\sum_{t=1}^{T} U(\boldsymbol{a}^{(t)}; \boldsymbol{\mu}, \boldsymbol{P})$. Since the system is stationary in $t$, the optimal action profile across different time slots can be expressed as:

$$\boldsymbol{a}^* \in \arg\max_{\boldsymbol{a} \in \mathcal{A}} U(\boldsymbol{a}; \boldsymbol{\mu}, \boldsymbol{P}). \tag{2}$$

where $\mathcal{A} \triangleq [M]^K$. Note that $\boldsymbol{a}^*$ is unknown to the decision maker because the parameters $\boldsymbol{\mu}$ and $\boldsymbol{P}$ are unknown. We define the regret as:

$$\text{Reg}_T \triangleq \mathbb{E}\left[\sum_{t=1}^{T} \left(U(\boldsymbol{a}^*; \boldsymbol{\mu}, \boldsymbol{P}) - U(\boldsymbol{a}^{(t)}; \boldsymbol{\mu}, \boldsymbol{P})\right)\right].$$

*Remark* 3.1. The utility function $U(\boldsymbol{a}^{(t)}; \boldsymbol{\mu}, \boldsymbol{P})$ has a nonlinear combinatorial structure with respect to $\boldsymbol{\mu}, \boldsymbol{P}$ and it has a cost term. As a consequence, arms with large per unit rewards are not necessarily favorable. There are in total $|\mathcal{A}| = M^K$ action profiles. Thus, locating $\boldsymbol{a}^*$ is nontrivial. Distinguishing optimal action profile from sub-optimal allocation from feedback is not easy. It is nontrivial to tackle this nonlinear combinatorial structure to reveal fundamental learning limits and balance the exploring vs. exploitation tradeoff.

## 4 FUNDAMENTAL LEARNING LIMITS

We reveal fundamental limits of learning the optimal action profile by proving instance independent and instance dependent regret lower bounds.

**Theorem 4.1.** *For any learning algorithm, there exists an instance of MSB-PRS such that*

$$Reg_T \geq \frac{1}{27}\alpha_1 \sigma \sqrt{MKT}.$$

*Furthermore, $Reg_T \geq \Omega(\alpha_1 \sigma \sqrt{KMT})$.*

The key proof idea is identifying one special instances of the MSB-PRS that are composed of carefully designed multiple independent groups classical multi-armed bandits and batched MP-MAB. For each group, we apply Theorem 15.2 of Lattimore & Szepesvári (2020) to bound its regret lower bound. Finally, summing them up across groups we obtain the instance independent regret lower bound.

**Theorem 4.2.** *Consider $\Delta \in \mathbb{R}_+$ utility gap MSB-PRS, i.e., the class of MSB-PRS satisfy*

$$U(\boldsymbol{a}^*; \boldsymbol{\mu}, \boldsymbol{P}) - \max_{\boldsymbol{a}:U(\boldsymbol{a};\boldsymbol{\mu},\boldsymbol{P})\neq U(\boldsymbol{a}^*;\boldsymbol{\mu},\boldsymbol{P})} U(\boldsymbol{a}; \boldsymbol{\mu}, \boldsymbol{P}) = \Delta.$$

*For any consistent play allocation algorithm, there exists $\Delta$ utility gap instances of MSB-PRS, such that the regret of any consistent learning algorithm on them satisfies*

$$\liminf_{T \to \infty} \frac{Reg_T}{\ln T} \geq \alpha_1 \sigma^2 \frac{M}{\Delta}\left(\frac{\ln K}{\ln T} + 1\right).$$

The idea of restricting to $\Delta$ utility gap MSB-PRS in proving the instance dependent regret lower bound follows the work Kveton et al. (2015a), which proves the instance dependent regret lower bound of stochastic combinatorial semi-bandits restricting to gap instances, instead of the basic model parameters. The proof routine is similar to that of the Theorem 4.1. The constructed special instances of the MSB-PRS are nearly the same as that of Theorem 4.1, except that for each group of bandits, we carefully design their mean gap, such that the total gap equals $\Delta$. We apply Theorem 16.2 of Lattimore & Szepesvári (2020) to bound the asymptotic lower bound. Finally, summing them up across groups we obtain the instance dependent regret lower bound.

## 5 Efficient Learning Algorithms

### 5.1 Efficient Computation Oracle

Given model parameters, we design `MSB-PRS-OffOpt` to locate the optimal action profile, which will serve as an efficient computation oracle for learning the optimal action profile.

**Bipartite graph formulation.** We formulate a complete weighted bipartite graph with node set $\mathcal{U} \cup \mathcal{V}$ and edge set $\mathcal{U} \times \mathcal{V}$, where $\mathcal{U} \cap \mathcal{V} = \emptyset$ and

$$\mathcal{U} \triangleq \{u_1, \ldots, u_K\}, \quad \mathcal{V} \triangleq \bigcup_{m \in [M]} \mathcal{V}_m, \; \mathcal{V}_m \triangleq \{v_{m,1}, \ldots, v_{m,K}\}.$$

The node $u_k \in \mathcal{U}$ corresponds to play $k \in [K]$. The node set $\mathcal{V}_m$ corresponds to arm $m \in [M]$. Nodes $v_{m,j} \in \mathcal{V}_m$, where $j \in [K]$, are designed to capture the prioritized resource sharing mechanism.

Denote $\Lambda_m(k, \ell)$ as the marginal utility contribution of play $k$ on an arm when it is ranked $\ell$-th among all plays pulling this arm, formally

$$\Lambda_m(k, \ell) \triangleq \alpha_k \mu_m P_{m,\ell} - c_{k,m}.$$

The $U_m(\boldsymbol{a}; \mu_m, \boldsymbol{P}_m)$ can be decomposed as:

$$U_m(\boldsymbol{a}; \mu_m, \boldsymbol{P}_m) = \sum_{k \in [K]} \mathbb{1}_{\{a_k = m\}} \Lambda_m(k, \ell_k(\boldsymbol{a})),$$

where $\ell_k(\boldsymbol{a})$ denote the rank of play on the arm $a_k$ according to the prioritized capacity sharing mechanism. Denote $\theta_k$ as the number of plays proceeding play $k$ with respect to their priority weights

$$\theta_k \triangleq |\{k' | \alpha_{k'} \geq \alpha_k\}|.$$

The prioritized capacity sharing mechanism implies an upper bound on the rank of $k$, i.e., $\ell_k(\boldsymbol{a}) \leq \theta_k$. Namely, on each arm, play $k$ would be ranked at most $\theta_k$-th regardless of the number of plays assigned to this arm.

Denote a weight function over the edge set as: $W : \mathcal{U} \times \mathcal{V} \to \mathbb{R}$. The weight of the edge $(u_k, v_{m,j})$ is defined as:

$$W(u_k, v_{m,j}) = \begin{cases} \Lambda_m(k, j), & \text{if } j \leq \theta_k, \\ -\infty, & \text{otherwise.} \end{cases}$$

The weight $W(u_k, v_{m,j})$ quantifies the marginal utility contribution of play $k$ for pulling arm $m$, when it is ranked $j$-th. As imposed by the prioritized capacity sharing mechanism, the rank of play $k$ can not exceed $\theta_k$. We thus set the utility associated with such invalid rank as $-\infty$ to disable these edges. Denote the weighted bipartite graph as $G = (\mathcal{U} \cup \mathcal{V}, \mathcal{U} \times \mathcal{V}, W)$.

**From action profiles to matchings**. Let $\mathcal{M} \subseteq \mathcal{U} \times \mathcal{V}$ denote a matching in graph $G$, which is a set of pairwise non-adjacent edges, i.e., $|\{u | (u, v) \in \mathcal{M}\}| = |\{v | (u, v) \in \mathcal{M}\}| = |\mathcal{M}|$. Denote the index of the arm that is linked to node $v_{m,j}$ under $\mathcal{M}$ as

$$\phi_{m,j}(\mathcal{M}) \triangleq \begin{cases} k, & \text{if } (u_k, v_{m,j}) \in \mathcal{M}, \\ 0, & \text{otherwise,} \end{cases}$$

where index $0$ is defined as a dummy play and we define its weight is as $\alpha_0 = 0$. Denote an indicator function associated with $\mathcal{M}$ as:

$$b_{m,j}(\mathcal{M}) \triangleq \begin{cases} 1, & \text{if } \exists k, (u_k, v_{m,j}) \in \mathcal{M}, \\ 0, & \text{otherwise.} \end{cases}$$

We next define a class of matchings that can be connected to the action profiles.

**Definition 5.1.** A matching $\mathcal{M}$ is: (1) $\mathcal{U}$-saturated if $\{u | (u, v) \in \mathcal{M}\} = \mathcal{U}$; (2) $\mathcal{V}$-monotone if $b_{m,j}(\mathcal{M}) \geq b_{m,j'}(\mathcal{M}), \forall j < j'$; (3) priority compatible if $\alpha_{\phi_{m,j}(\mathcal{M})} \geq \alpha_{\phi_{m,j'}(\mathcal{M})}, \forall j < j'$.

The $\mathcal{U}$-saturated property states that each play node is an endpoint of one edge of $\mathcal{M}$. The $\mathcal{V}$-monotone property states that end points of $\mathcal{M}$ on the $\mathcal{V}_m$ side forms an increasing set, i.e., it can be expressed as $\{v_{m,1}, \ldots, v_{m,J}\}$, where $J = |\{v | (u, v) \in \mathcal{M}\} \cap \mathcal{V}_m|$.

**Lemma 5.2.** *Action profile $\boldsymbol{a} \in \mathcal{A}$ can be mapped into a $\mathcal{U}$-saturated, $\mathcal{V}$-monotone, and priority compatible matching $\widetilde{\mathcal{M}}(\boldsymbol{a}) = \{(u_k, v_{a_k, \ell_k(\boldsymbol{a})}) | k \in [K]\}$. Furthermore, it holds that $U(\boldsymbol{a}; \boldsymbol{\mu}, \boldsymbol{P}) = \sum_{(u,v) \in \widetilde{\mathcal{M}}(\boldsymbol{a})} W(u, v)$, and $\widetilde{\mathcal{M}}(\boldsymbol{a}) \neq \widetilde{\mathcal{M}}(\boldsymbol{a}')$ for any $\boldsymbol{a} \neq \boldsymbol{a}'$.*

Lemma 5.2 states that each action profile can be mapped into a $\mathcal{U}$-saturated , $\mathcal{V}$-monotone and priority compatible matching with utility equals the weights of the matching.

**From matchings to action profiles.** In the following lemma, we show that a $\mathcal{U}$-saturated, $\mathcal{V}$-monotone, and priority compatible matching can be mapped into an action profile with weights of the matching equals the utility.

**Lemma 5.3.** *A $\mathcal{U}$-saturated, $\mathcal{V}$-monotone, and priority compatible matching $\mathcal{M}$ can be mapped into action profile $\widetilde{\boldsymbol{a}}(\mathcal{M}) \triangleq (\widetilde{a}_k(\mathcal{M}) : \forall k \in [K])$, where*

$$\widetilde{a}_k(\mathcal{M}) = \sum_{m \in [M]} m \sum_{j \in [K]} \mathbb{1}_{\{\phi_{m,j}(\mathcal{M})=k\}}.$$

*Furthermore, $U(\widetilde{\boldsymbol{a}}(\mathcal{M}), \boldsymbol{\mu}, \boldsymbol{P}) = \sum_{(u,v) \in \mathcal{M}} W(u,v)$.*

**Locating the optimal action profile.** Lemma 5.2 and 5.3 imply that locating the optimal action profile is equivalent to searching the $\mathcal{U}$-saturated, $\mathcal{V}$-monotone, and priority compatible matching with the maximum total weights. However, a maximum weighted matching may not be $\mathcal{U}$-saturated, $\mathcal{V}$-monotone and priority compatible. This hinders one to apply the maximum weighted matching algorithm. For any $\mathcal{U}$-saturated matching $\mathcal{M}$, if it is not $\mathcal{V}$-monotone or priority compatible, it can be adjusted to be a $\mathcal{V}$-monotone and priority compatible matching $\mathcal{M}'$:

$$\mathcal{M}' \triangleq \bigcup_{m \in [M]} \bigcup_{j=1}^{|\mathcal{K}_m|} \{(u_{L_{m,j}}, v_{m,j})\} \tag{3}$$

where $\mathcal{K}_m = \{j | \phi_{m,j}(\mathcal{M}) \neq 0\}$ denotes a set of plays linked to arm $m$ by $\mathcal{M}$, $L_{m,1}, \ldots, L_{m,|\mathcal{K}_m|}$ is a ranked list of $\mathcal{K}_m$ such that $L_{m,j} < L_{m,j'}, \forall j < j'$. Furthermore, it can be easily verified that $\sum_{(u,v) \in \mathcal{M}} W(u,v) \leq \sum_{(u,v) \in \mathcal{M}'} W(u,v)$. The implication is that one can first locate the maximum weighted matching (the maximum weighted matching is $\mathcal{U}$-saturated). If it does not have all three desired properties, one can apply the above strategy to adjust it to have all three desired properties. Locating the maximum weight matching is a well studied problem. The Hungarian algorithm and its variants such as Crouse *et al. Crouse (2016)* provide computationally efficient algorithms for this problem. Algorithm 1 combines the above elements to locate the optimal action profile. The essential computational complexity is the maximum weighted matching. The computational complexity of Algorithm 1 is $O(M^3 K^3)$, if Crouse *et al. Crouse (2016)* is applied.

---

**Algorithm 1** MSB-PRS-OffOpt $(\boldsymbol{\mu}, \boldsymbol{P})$

---

1: $G \leftarrow (\mathcal{U} \cup \mathcal{V}, \mathcal{U} \times \mathcal{V}, W)$
2: $\mathcal{M} \leftarrow \texttt{MaximumWeightedMatching(G)}$
3: If $\mathcal{M}$ does not have three desired properties, adjust it according to Eq. (3)
4: $\widetilde{a}_k(\mathcal{M}) \leftarrow \sum_{m \in [M]} m \sum_{j \in [K]} \mathbb{1}_{\{\phi_{m,j}(\mathcal{M})=k\}}$
5: **Return:** $\widetilde{\boldsymbol{a}}(\mathcal{M}) = [\widetilde{a}_k(\mathcal{M}) : k \in [K]]$

---

### 5.2 Efficient Learning Algorithm

**Approximate UCB based algorithm.** Note that in time slot $t+1$, the decision maker has access to the historical feedback up to time slot $t$, formally $\mathcal{H}_t \triangleq (\boldsymbol{D}^{(1)}, \boldsymbol{X}^{(1)}, \boldsymbol{a}^{(1)}, \ldots, \boldsymbol{D}^{(t)}, \boldsymbol{X}^{(t)}, \boldsymbol{a}^{(t)})$. Denote the complementary cumulative probability matrix estimated from $\mathcal{H}_t$ as $\widehat{\boldsymbol{P}}^{(t)} \triangleq [\widehat{P}_{m,d}^{(t)} : m \in [M], d \in [d_{max}]]$, where the $\widehat{P}_{m,d}^{(t)}$ is the empirical average:

$$\widehat{P}_{m,d}^{(t)} \triangleq \frac{\sum_{s=1}^{t} \mathbb{1}_{\{N_m^{(s)} \geq 1\}} \mathbb{1}_{\{D_m^{(s)} \geq d\}}}{\sum_{s=1}^{t} \mathbb{1}_{\{N_m^{(s)} \geq 1\}}}. \tag{4}$$

Denote the mean vector estimated from $\mathcal{H}_t$ as $\widehat{\boldsymbol{\mu}}^{(t)} = [\widehat{\mu}_m^{(t)} : m \in [M]]$, where the $\widehat{\mu}_m^{(t)}$ is the empirical average:

$$\widehat{\mu}_m^{(t)} \triangleq \frac{\sum_{s=1}^{t} \sum_{k=1}^{K} \mathbb{1}_{\{X_k^{(s)} \neq \text{null}\}} \mathbb{1}_{\{a_k^{(s)}=m\}} X_k^{(s)} / \alpha_k}{\sum_{s=1}^{t} \sum_{k=1}^{K} \mathbb{1}_{\{X_k^{(s)} \neq \text{null}\}} \mathbb{1}_{\{a_k^{(s)}=m\}}}. \tag{5}$$

The following lemma states a confidence band for the above estimators.

**Lemma 5.4.** *The estimators $\widehat{P}_{m,d}^{(t)}$ and $\widehat{\mu}_m^{(t)}$ satisfy:*

$$\mathbb{P}\left[\exists t, m, |\mu_m - \widehat{\mu}_m^{(t)}| \geq \epsilon_m^{(t)}\right] \leq 2M\delta, \ \mathbb{P}\left[\exists t, m, d, |\widehat{P}_{m,d}^{(t)} - P_{m,d}| \geq \lambda_m^{(t)}\right] \leq 2Md_{\max}\delta,$$

*where $\delta \in (0,1)$, $\epsilon_m^{(t)}$ and $\lambda_m^{(t)}$ are derived as*

$$\epsilon_m^{(t)} = \begin{cases} \sqrt{2\sigma^2(\widetilde{n}_m^{(t)}+1)\ln\frac{\sqrt{\widetilde{n}_m^{(t)}+1}}{\delta}\frac{1}{\widetilde{n}_m^{(t)}}}, & \text{if } \widetilde{n}_m^{(t)} \geq 1, \\[2ex] +\infty, & \text{if } \widetilde{n}_m^{(t)} = 0, \end{cases}$$

$$\lambda_m^{(t)} = \begin{cases} \sqrt{\frac{n_m^{(t)}+1}{2}\ln\frac{\sqrt{n_m^{(t)}+1}}{\delta}\frac{1}{n_m^{(t)}}} \wedge 1, & \text{if } n_m^{(t)} \geq 1, \\[2ex] 1, & \text{if } n_m^{(t)} = 0, \end{cases}$$

*where the operation $\wedge$ means selecting the smaller value between two, $\widetilde{n}_m^{(t)} = \sum_{s=1}^t \sum_{k=1}^K \mathbb{1}_{\{X_k^{(s)} \neq null\}} \mathbb{1}_{\{a_k^{(s)} = m\}}$ and $n_m^{(t)} = \sum_{s=1}^t \mathbb{1}_{\{N_m^{(t)} \geq 1\}}$.*

For simplicity, we denote $\boldsymbol{\epsilon}^{(t)} = [\epsilon_m^{(t)} : m \in [M]]$ and $\boldsymbol{\lambda}^{(t)} = [\lambda_m^{(t)} : m \in [M]]$. Based on the above lemma, the exact UCB index of action profile $\boldsymbol{a}$ can be expressed as:

$$\text{Exact-UCB}^{(t)}(\boldsymbol{a}) = \max_{\substack{\boldsymbol{\mu},\boldsymbol{P}, |\widehat{\mu}_m^{(t)} - \mu_m| \leq \epsilon_m^{(t)}, \forall m \\ |\widehat{P}_{m,d}^{(t)} - P_{m,d}| \leq \lambda_m^{(t)}, \forall m, d}} U(\boldsymbol{a}, \boldsymbol{\mu}, \boldsymbol{P}).$$

The Exact-UCB$^{(t)}(\boldsymbol{a})$ has a potential computational issue in locating the action profile with larger index. Specifically, the Exact-UCB$^{(t)}(\boldsymbol{a})$ may attain the max value at different selections of $\boldsymbol{\mu}, \boldsymbol{P}$ for different action profiles, especially when the confidence band fails. In this case, to locate the action profile one can only resort to exhaustive search, resulting in a computational complexity of $O(K^M)$. To avoid this problem, we propose to use the approximate UCB index:

$$\text{UCB}^{(t)}(\boldsymbol{a}) = U(\boldsymbol{a}, \widehat{\boldsymbol{\mu}}^{(t)} + \boldsymbol{\epsilon}^{(t)}, \widehat{\boldsymbol{P}}^{(t)} + \boldsymbol{\lambda}^{(t)}). \tag{6}$$

One advantage of UCB$^{(t)}(\boldsymbol{a})$ over Exact-UCB$^{(t)}(\boldsymbol{a})$ is that all action profile share the same parameter $\widehat{\boldsymbol{\mu}}^{(t)} + \boldsymbol{\epsilon}^{(t)}, \widehat{\boldsymbol{P}}^{(t)} + \boldsymbol{\lambda}^{(t)}$. Algorithm 1 locates the action profile attaining the maximum UCB$^{(t)}(\boldsymbol{a})$ with a computational complexity of $O(K^3M^3)$. As we shown in the proof of instance independent upper bound, the monotonicity of utility function with respect to $\boldsymbol{\mu}$ and $\boldsymbol{P}$ element-wisely guarantees the UCB validity of UCB$^{(t)}(\boldsymbol{a})$. The action profile in round $t$ is then selected by:

$$\boldsymbol{a}^{(t)} \in \arg\max_{\boldsymbol{a} \in \mathcal{A}} \text{UCB}^{(t-1)}(\boldsymbol{a}).$$

Summarizing the above ideas together, Algorithm 2 outlines an approximate UCB based algorithm.

---

**Algorithm 2** MSB-PRS-ApUCB ( $\mathcal{H}_t$ )

---

1: $\widehat{P}_{m,d}^{(0)} \leftarrow 1, \widehat{\mu}_m^{(0)} \leftarrow 0$
2: **for** $t = 1, \ldots, T$ **do**
3:     Calculate $\epsilon_m^{(t-1)}$ and $\lambda_m^{(t-1)}$ applying Lemma (5.4)
4:     $\boldsymbol{a}^{(t)} \leftarrow$ MSB-PRS-OffOpt($\widehat{\boldsymbol{\mu}}^{(t-1)} + \boldsymbol{\epsilon}^{(t-1)}, \widehat{\boldsymbol{P}}^{(t-1)} + \boldsymbol{\lambda}^{(t-1)}$)
5:     Observe $\boldsymbol{D}^{(t)}$ and $\boldsymbol{X}^{(t)}$
6:     Update $\widehat{P}_{m,d}^{(t)}$ via Eq. (4), $\forall m \in \{m' | N_{m'}^{(t)} > 0\}$
7:     Update $\widehat{\mu}_m^{(t)}$ via Eq. (5), $\forall m \in \{m' | N_{m'}^{(t)} > 0\}$
8: **end for**

---

**Regret upper bounds.** The following two theorems state the instance independent and instance dependent regret lower bound individually.

**Theorem 5.5.** *The instance independent regret upper bound of Algorithm 2 can be derived as:*

$$Reg_T \leq 2M(1 + d_{\max})K\mu_{\max} + 8\alpha_1(\mu_{\max} + 1)KM\sqrt{T}\left(\sqrt{\ln T} + 4\sigma\sqrt{\ln KT}\sqrt{K/M}\right)$$

*Furthermore, $Reg_T \leq O(\alpha_1 \sigma \mu_{max}\sqrt{KMT}K\sqrt{\ln KT})$.*

Compared to the instance independent regret lower bound derived in Theorem 4.1, the regret upper bound matches the lower bound up to a factor of $K\sqrt{\ln KT}$. The key proof idea is via exploiting the monotone property of the utility function to prove the validity of the approximate UCB index.

**Theorem 5.6.** *The instance dependent regret upper bound of Algorithm 2 can be derived as:*

$$Reg_T \leq 96MK^2\alpha_1^2(2\sigma + 1)^2\frac{1}{\Delta}\ln KT + 2M(1 + d_{\max})K\mu_{\max}$$

*Furthermore, $Reg_T \leq O(MK^2\alpha_1^2\sigma^2\frac{1}{\Delta}\ln KT)$.*

Compared to the instance dependent regret lower bound derived in Theorem 4.2, the regret upper bound matches the lower bound up to a factor of $\alpha_1 K^2$. The key proof idea of tackling the aforementioned nonlinear combinatorial structure in the proof is via exploiting the monotone property of the utility function to show suboptimal allocations make progress in improving the estimation accuracy of poor estimated parameters, which gear the learning algorithm toward identifying more favorable suboptimal allocations. Furthermore, group suboptimal action profiles with respect to their gap to the optimal action profile, with a double trick on determining the desired gap for each group.

**Discussion on tightness.** We believe that closing the regret gap is an open problem, since MSB-PRS is neither a standard MP-MAB model nor a standard combinatorial bandit model.

# 6 SYNTHETIC EXPERIMENTS

**Parameter setting**. We consider $M = 5$ arms and $K = 10$ plays. It is essential to note that we will systematically vary $M$ and K to assess the performance of our proposed algorithm. The probability mass function and the reward distribution is same as Chen *et al.* Chen & Xie (2022). We designate the movement cost as $c_{k,m} = \eta|(k \mod M) - m|/\max\{K, M\}$, where $\eta \in \mathbb{R}_+$ is a hyper-parameter that controls the scale of the cost. Unless explicitly varied, we adopt the following default parameters: $T = 10^4$, $\delta = 1/T$, $K = 10$ plays, $M = 5$ arms, $\eta = 1$, $\sigma = 0.2$ and the U-Shape reward. Furthermore, the weight of half of plays is 3 and the other half is 1. We consider two baselines: (1) `OnlinActPrf` Chen & Xie (2022), which considers the setting with expert feedback on capacity and homogeneous plays without priority capacity sharing; (2) `OnlinActPrf-v`, which is a variant of `OnlinActPrf` enabling UCB on the capacity distribution estimation. Due to page limit, more details on the setting are in appendix.

**Impact of the number of arms** We varied the number of arms, denoted as $M$, across three settings: $M = 5$, 10, and 15, and plotted the regret of three algorithms. In Fig. 1a, it is evident that the regret curves for `MSB-PRS-ApUCB` under $M = 5$, 10, and 20 initially exhibit a sharp increase before leveling off, indicating a sub-linear regret. Additionally, one can find that the convergence rate of `MSB-PRS-ApUCB` regret gradually decreases with an increase in $M$. Fig. 1b illustrates that the regret curves for `OnlinActPrf` and `OnlinActPrf-v` follow a linear trend, while the regret curve for `MSB-PRS-ApUCB` consistently remains at the bottom. This observation confirms that `MSB-PRS-ApUCB` yields the smallest regret compared to the two baseline algorithms. This trend persists even when $M = 10$ and 15, as shown in Fig. 1c and 1d, respectively. Due to page limit, more experiments are presented in the appendix.

**Impact of the number of plays** We varied the number of plays, denoted as $K$, across three settings: $K = 10$, 15, and 20, and plotted the regret of three algorithms. In Fig. 2a, it is evident that the regret curves for `MSB-PRS-ApUCB` under $K = 10$, 15, and 20 initially exhibit a sharp increase before plateauing, indicating a sub-linear regret. Additionally, Fig. 2b illustrates that the regret curves for `OnlinActPrf` and `OnlinActPrf-v` follow a linear trend, while the regret curve for `MSB-PRS-ApUCB` consistently remains at the bottom. This observation confirms that `MSB-PRS-ApUCB` yields the smallest regret compared to the two baseline algorithms. This trend persists even when $K = 15$ and 20, as shown in Fig. 2c and 2d, respectively.

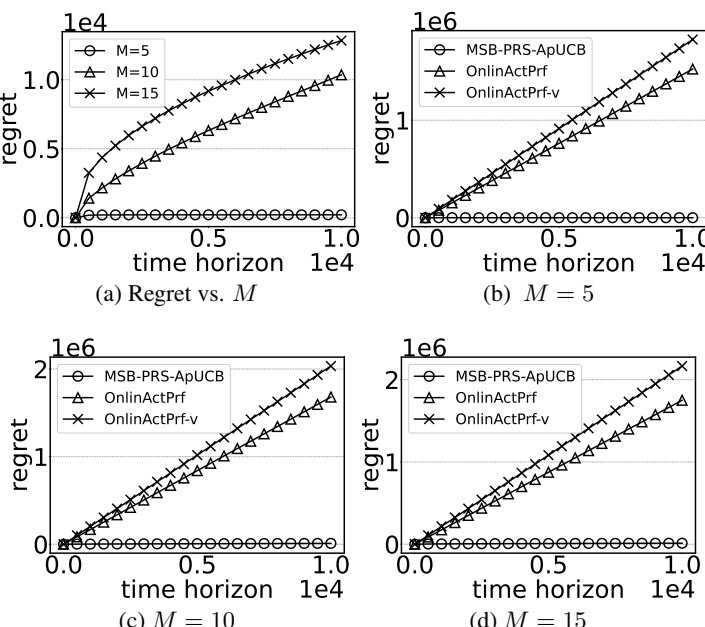

Figure 1: Impact of Number of Arms.

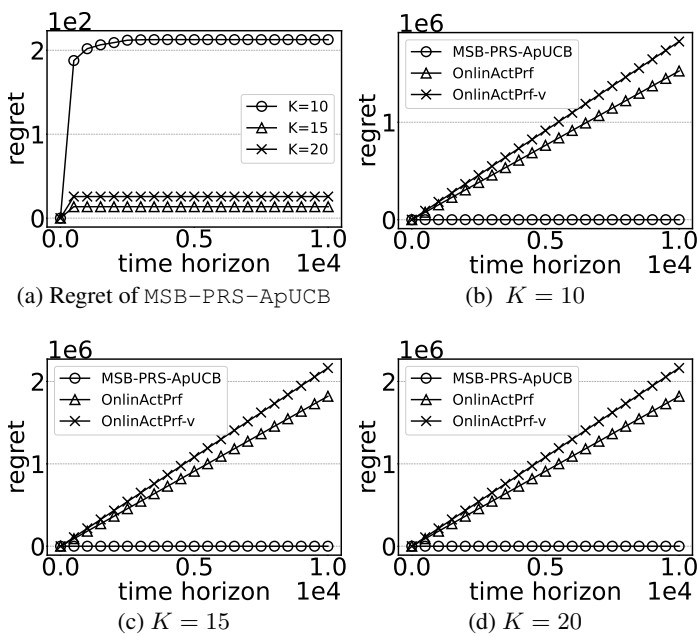

Figure 2: Impact of Number of plays.

## 7 CONCLUSION

This paper proposes MSB-PRS. An algorithm is designed to locate the optimal play allocation policy with a complexity of $O(M^3 K^3)$. Instance independent and instance dependent regret lower bounds of $\Omega(\alpha_1 \sigma \sqrt{KMT})$ and $\Omega(\alpha_1 \sigma^2 \frac{M}{\Delta} \ln T)$ are proved respectively. An approximate UCB based algorithm is designed which has a per round computational complexity of $O(M^3 K^3)$ and has sublinear independent and dependent regret upper bounds matching the corresponding lower bounds up to acceptable factors.

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

APPENDIX

This appendix contains two section. Section A presents technical proofs to lemmas and theorems Section B presents additional experiments.

## A  TECHNICAL PROOFS

### A.1  PROOF OF LEMMA 5.2

We first prove the $\mathcal{U}$-saturated property. Note that each action profile $\boldsymbol{a}$ assigns all plays to arms and each arm is assigned to only one arm. Thus, it holds that $\{u|(u,v) \in \widetilde{\mathcal{M}}(\boldsymbol{a})\} = \mathcal{U}$.

Now we prove the $\mathcal{V}$-monotone and priority compatible property. Note that all the plays assigned to an arm are ordered based on $\ell_k(\boldsymbol{a})$. This order list is monotone and priority compatible. Thus, $\widetilde{\mathcal{M}}(\boldsymbol{a})$ is monotone and priority compatible.

We prove the utility preserving property. Let $\mathcal{J}_m = \{j|a_j = m\}$ denote the set of all plays that pull arm $m$. By the monotone and priority compatible property of $\ell_k(\boldsymbol{a})$ and some basic arguments we have:

$$
\begin{aligned}
U(\boldsymbol{a}; \boldsymbol{\mu}, \boldsymbol{P}) &= \sum_{m \in [M]} U_m(\boldsymbol{a}; \mu_m, \boldsymbol{P}_m) \\
&= \sum_{m \in [M]} \sum_{k \in [K]} \mathbb{1}_{\{a_k = m\}} \Lambda_m(k, \ell_k(\boldsymbol{a})) \\
&= \sum_{m \in [M]} \sum_{k \in [K]} \mathbb{1}_{\{a_k = m\}} W(u_k, v_{m, \ell_k(\boldsymbol{a})}) \\
&= \sum_{m \in [M]} \sum_{k \in [K]} \mathbb{1}_{\{\phi_{m, \ell_k(\boldsymbol{a})} = k\}} W(u_k, v_{m, \ell_k(\boldsymbol{a})}) \\
&= \sum_{m \in [M]} \sum_{k \in [K]} \sum_{j=1}^{|\mathcal{J}_m|} \mathbb{1}_{\{\phi_{m,j} = k\}} W(u_k, v_{m,j}) \\
&= \sum_{m \in [M]} \sum_{j=1}^{|\mathcal{J}_m|} \sum_{k \in [K]} \mathbb{1}_{\{\phi_{m,j} = k\}} W(u_k, v_{m,j}) \\
&= \sum_{m \in [M]} \sum_{j=1}^{|\mathcal{J}_m|} W(u_k, v_{m,j}) = \sum_{(u,v) \in \widetilde{\mathcal{M}}(\boldsymbol{a})} W(u, v).
\end{aligned}
$$

Finally, we prove the uniqueness. Consider two action profiles $\boldsymbol{a}$ and $\boldsymbol{a}'$ such that $\boldsymbol{a} \neq \boldsymbol{a}'$. Then, we know that there exists at least one element that $\boldsymbol{a}$ and $\boldsymbol{a}'$ do not agree. This means that there exists at least one play that pulls different arms under $\boldsymbol{a}$ and $\boldsymbol{a}'$ respectively. Then, we conclude the uniqueness. This proof is then complete.

### A.2  PROOF OF LEMMA 5.3

One can easily verify that $\widetilde{a}_k(\mathcal{M})$ is the arm pulled by play $k$ under the matching $\mathcal{M}$. Since the matching $\mathcal{M}$ is $\mathcal{U}$-saturated, each play is assigned to one arm. Namely, $\widetilde{\boldsymbol{a}}(\mathcal{M})$ forms a valid assignment. The remaining thing is to show the the utility of $\widetilde{\boldsymbol{a}}(\mathcal{M})$ matches the sum of weights of $\mathcal{M}$. Let $\widetilde{\mathcal{V}}_m$ denote the end points of $\mathcal{M}$ on the arm node side that belong to the set $\mathcal{V}_m$, formally

$$
\widetilde{\mathcal{V}}_m = \{v|(u,v) \in \mathcal{M}\} \cap \mathcal{V}_m.
$$

Note that the matching $\mathcal{M}$ is $\mathcal{V}$-monotone. This means that $\widetilde{\mathcal{V}}_m$ can be expressed as $\widetilde{\mathcal{V}}_m = \{v_{m,1}, \ldots, v_{m,|\widetilde{\mathcal{V}}_m|}\}$. Furthermore, $(u_{\phi_{m,j}(\mathcal{M})}, v_{m,j})$ is an edge of the matching with the arm side

end point in the set $\widetilde{\mathcal{V}}_m$. The total weights of the matching $\mathcal{M}$ can be decomposed as

$$\sum_{(u,v)\in\mathcal{M}} W(u,v) = \sum_{m\in[M]} \sum_{j=1}^{|\widetilde{\mathcal{V}}_m|} W(u_{\phi_{m,j}(\mathcal{M})}, v_{m,j}) = \sum_{m\in[M]} \sum_{j=1}^{|\widetilde{\mathcal{V}}_m|} \Lambda_m(\phi_{m,j}(\mathcal{M}), j).$$

We next complete this proof by showing that

$$\sum_{j=1}^{|\widetilde{\mathcal{V}}_m|} \Lambda_m(\phi_{m,j}(\mathcal{M}), j) = U_m(\widetilde{\boldsymbol{a}}(\mathcal{M}); \mu_m, \boldsymbol{P}_m).$$

The matching $\mathcal{M}$ being priority compatible implies that

$$\ell_{\phi_{m,j}(\mathcal{M})}(\widetilde{\boldsymbol{a}}(\mathcal{M})) = j.$$

Then, it follows that:

$$\sum_{j=1}^{|\widetilde{\mathcal{V}}_m|} \Lambda_m(\phi_{m,j}(\mathcal{M}), j) = \sum_{j=1}^{|\widetilde{\mathcal{V}}_m|} \Lambda_m(\phi_{m,j}(\mathcal{M}), \ell_{\phi_{m,j}(\mathcal{M})}(\widetilde{\boldsymbol{a}}(\mathcal{M})))$$

$$= \sum_{j=1}^{|\widetilde{\mathcal{V}}_m|} \sum_{k\in[K]} \mathbb{1}_{\{\phi_{m,j}(\mathcal{M})=k\}} \Lambda_m(k, \ell_k(\widetilde{\boldsymbol{a}}(\mathcal{M})))$$

$$= \sum_{k\in[K]} \sum_{j=1}^{|\widetilde{\mathcal{V}}_m|} \mathbb{1}_{\{\phi_{m,j}(\mathcal{M})=k\}} \Lambda_m(k, \ell_k(\widetilde{\boldsymbol{a}}(\mathcal{M})))$$

$$= \sum_{k\in[K]} \mathbb{1}_{\{\widetilde{a}_k(\mathcal{M})=m\}} \Lambda_m(k, \ell_k(\widetilde{\boldsymbol{a}}(\mathcal{M})))$$

$$= U_m(\widetilde{\boldsymbol{a}}(\mathcal{M}); \mu_m, \boldsymbol{P}_m),$$

This proof is then complete.

### A.3 PROOF OF LEMMA 5.4

Note that the range of $\mathbb{1}_{\{D_m^{(s)} \geq d\}}$ is within $\{0, 1\}$. It is a $\frac{1}{2}$-subgaussian random variable. Note that $D_m^{(t)}$'s are independent across $t$ and $m$. Lemma 10 of Maillard (2017) straightforwardly yields

$$\mathbb{P}\left[\exists t, |\widehat{P}_{m,d}^{(t)} - P_{m,d}| \geq \sqrt{\frac{n_m^{(t)}+1}{2} \ln \frac{\sqrt{n_m^{(t)}+1}}{\delta} \frac{1}{n_m^{(t)}}}\right] \leq 2\delta.$$

Note that $P_{m,d} \in [0,1]$ and $\widehat{P}_{m,d}^{(t)} \geq 0$. Thus the above inequality can be made into

$$\mathbb{P}\left[\exists t, |\widehat{P}_{m,d}^{(t)} - P_{m,d}| \geq \sqrt{\frac{n_m^{(t)}+1}{2} \ln \frac{\sqrt{n_m^{(t)}+1}}{\delta} \frac{1}{n_m^{(t)}}} \wedge 1\right] \leq 2\delta.$$

This is equivalent to

$$\mathbb{P}\left[\exists t, |\widehat{P}_{m,d}^{(t)} - P_{m,d}| \geq \lambda_m^{(t)}\right] \leq 2\delta.$$

The union bounds implies that

$$\mathbb{P}\left[\exists t, m, d, |\widehat{P}_{m,d}^{(t)} - P_{m,d}| \geq \lambda_m^{(t)}\right] \leq \sum_{m\in[M], d\in[d_{\max}]} \mathbb{P}\left[\exists t, |\widehat{P}_{m,d}^{(t)} - P_{m,d}| \geq \lambda_m^{(t)}\right] \leq 2M d_{\max}\delta.$$

By a similar argument, one can prove the confidence band with respect to the estimator $\widehat{\mu}_m^{(t)}$. This proof is then complete.

### A.4  PROOF OF THEOREM 4.1

This proof relies on the construction of a special instance of our model. The special instance is constructed as follows. The move cost of each arm is fixed to zero, formally

$$c_{k,m} = 0, \forall k, m.$$

All arms have the same weights, i.e.,

$$\alpha_1 = \alpha_1 = \cdots = \alpha_K.$$

The reward of each follows a Normal distribution:

$$R_m \sim \mathcal{N}(\mu_m, \sigma).$$

In each round, each arm generates $K$ units of resource, and the number of resource is deterministic, i.e.,

$$d_{max} = K, P_{m,d} = 1, \forall m, d.$$

Suppose that the reward mean satisfies:

$$\mu_1 > \mu_2 > \ldots > \mu_M.$$

Then the optimal action profile is assigning all plays to arm $m = 1$, formally

$$a_k^* = 1, \forall k.$$

Now we prove the regret lower bound based on the above constructed special instance of our problem. In each round the decision maker needs to assign $K$ plays to arms, and multiple arms can be assigned to the same arm. For each assigned play $a_k^{(t)}$, if it hits the optimal arm, i.e., $a_k^{(t)} = a_k^*$, it does not incur regret. If it hit a suboptimal arm, say $a_k^{(t)} = m$, where $m \neq 1$, it incurs an expected regret of $\mu_1 - \mu_m$. Thus in each round, the action profile $\boldsymbol{a}^{(t)}$ incurs an expected regret of

$$\sum_{k \in [K]} \alpha_1(\mu_1 - \mu_{a_k^{(t)}}).$$

Note that this is not MP-MAB, as multiple plays is allowed be assigned to the same arm and generates multiple independent rewards with the same distribution associated with this arm. We construct a relaxed variant of this problem to make it easier for learning. Four model, though the decision maker received $K$ rewards, but these reward can be used to update the decision only when this round ends. Namely, the decision maker can only update the decision when a round ends. To make the learning easier, we allow the decision maker to update the decision within each round, when new rewards is received. Specifically, we treat each round as $K$ sub-rounds. Each sub-round receives one reward and the decision maker can update the decision in each sub-round. This relaxation improves the data utilization and making the learning easier. To unify, we treat each sub-round as a real round. This leads to that the decision maker needs to assign one play in each round, but in total the decision maker needs to play $KT$ rounds. In each round of play, the decision maker observes all historical rewards before this round. Then this becomes a tradition $M$-armed bandit problem with $KT$ rounds of play. Applying Theorem 15.2 of Lattimore & Szepesvári (2020), this problem has a regret lower bound of

$$\frac{1}{27}\alpha_1 \sigma \sqrt{MKT}.$$

This proof is then complete.

### A.5  PROOF OF THEOREM 4.2

We construct the same instance of MSB-PRS as that in the proof of Theorem 4.1, except that we adding more elements to its reward mean. The reward mean satisfies:

$$\mu_2 = \mu_1 - \Delta/\alpha_1, \ldots, \mu_M = \mu_1 - \Delta/\alpha_1.$$

Note that the optimal assign is $a_k^* = 1$, namely

$$U(\boldsymbol{a}^*; \boldsymbol{\mu}, \boldsymbol{P}) = K\alpha_1\mu_1.$$

And the most favorable sub-optimal action profiles are the ones that only one play misses the arm $m = 1$, and all other plays hit the arm $m = 1$. This implies that

$$\max_{\boldsymbol{a}:U(\boldsymbol{a};\boldsymbol{\mu},\boldsymbol{P})\neq U(\boldsymbol{a}^*;\boldsymbol{\mu},\boldsymbol{P})} U(\boldsymbol{a};\boldsymbol{\mu},\boldsymbol{P}) = K\alpha_1\mu_1 - \Delta.$$

Thus we have

$$U(\boldsymbol{a}^*;\boldsymbol{\mu},\boldsymbol{P}) - \max_{\boldsymbol{a}:U(\boldsymbol{a};\boldsymbol{\mu},\boldsymbol{P})\neq U(\boldsymbol{a}^*;\boldsymbol{\mu},\boldsymbol{P})} U(\boldsymbol{a};\boldsymbol{\mu},\boldsymbol{P}) = \Delta.$$

With a similar argument of the proof of Theorem 4.1, we only need to analyze the relaxed variant of the instance. Note that the relaxed variant is a tradition $M$-armed bandit problem with $KT$ rounds of play. Applying Theorem 16.2 of Lattimore & Szepesvári (2020), the asymptotic regret lower bound of this variant is

$$2\alpha_1\sigma^2 \frac{M}{\Delta} \ln KT.$$

This proof is then complete.

### A.6  PROOF OF THEOREM 5.5

We decompose the proof into the following four steps.

**Step I:** Prove the confidence level of the approximate UCB index. We aim to prove the following holds

$$\mathbb{P}\left[\begin{array}{c}\forall t, \boldsymbol{a}, \text{UCB}^{(t)}(\boldsymbol{a}) - U(\boldsymbol{a};\boldsymbol{\mu},\boldsymbol{P}) \leq \sum_{m\in[M]} 4\alpha_1(\mu_{\max}+1)(\lambda_m^{(t)} + 2\epsilon_m^{(t)})\sum_{k\in[K]} \mathbb{1}_{\{a_k=m\}}, \\ \text{UCB}^{(t)}(\boldsymbol{a}) \geq U(\boldsymbol{a};\boldsymbol{\mu},\boldsymbol{P})\end{array}\right] \geq 1 - 2M(1+d_{\max})\delta.$$

Lemma 5.4 and union bounds imply:

$$\mathbb{P}\left[\forall t, m, d, |\mu_m - \widehat{\mu}_m^{(t)}| \leq \epsilon_m^{(t)}, |\widehat{P}_{m,d}^{(t)} - P_{m,d}| \leq \lambda_m^{(t)}\right] \geq 1 - 2M(1+d_{\max})\delta.$$

We next derive an upper bound of $\text{UCB}^{(t)}(\boldsymbol{a}) - U(\boldsymbol{a};\boldsymbol{\mu},\boldsymbol{P})$, given $|\mu_m - \widehat{\mu}_m^{(t)}| \leq \epsilon_m^{(t)}$ and $|\widehat{P}_{m,d}^{(t)} - P_{m,d}| \leq \lambda_m^{(t)}$. First we have:

$$\begin{aligned}\text{UCB}^{(t)}(\boldsymbol{a}) - U(\boldsymbol{a};\boldsymbol{\mu},\boldsymbol{P}) &= U(\boldsymbol{a},\widehat{\boldsymbol{\mu}}^{(t)}+\boldsymbol{\epsilon}^{(t)}, \widehat{\boldsymbol{P}}^{(t)}+\boldsymbol{\lambda}^{(t)}) - U(\boldsymbol{a};\boldsymbol{\mu},\boldsymbol{P}) \\ &= \sum_{m\in[M]} (U_m(\boldsymbol{a},\widehat{\mu}_m^{(t)}+\epsilon_m^{(t)}, \widehat{\boldsymbol{P}}_m^{(t)}+\boldsymbol{\lambda}_m^{(t)}) - U_m(\boldsymbol{a};\mu_m,\boldsymbol{P}_m)) \\ &= \sum_{m\in[M]} \Phi_m^{(t)},\end{aligned}$$

where $\Phi_m^{(t)}$ is defined as $\Phi_m^{(t)} \triangleq U_m(\boldsymbol{a},\widehat{\mu}_m^{(t)}+\epsilon_m^{(t)}, \widehat{\boldsymbol{P}}_m^{(t)}+\boldsymbol{\lambda}_m^{(t)}) - U_m(\boldsymbol{a};\mu_m,\boldsymbol{P}_m)$. Note that

$$U_m(\boldsymbol{a},\widehat{\mu}_m^{(t)} + \epsilon_m^{(t)}, \widehat{\boldsymbol{P}}_m^{(t)}+\boldsymbol{\lambda}_m^{(t)}) = (\widehat{\mu}_m^{(t)} + \epsilon_m^{(t)})\sum_{k\in[K]} \mathbb{1}_{\{a_k=m\}}\alpha_k(\widehat{P}_{m,\ell_k^{(t)}}^{(t)} + \lambda_m^{(t)}) - \sum_{k\in[M]} c_{k,m}\mathbb{1}_{\{a_k=m\}},$$

$$U_m(\boldsymbol{a};\mu_m,\boldsymbol{P}_m) = \mu_m \sum_{k\in[K]} \mathbb{1}_{\{a_k=m\}}\alpha_k P_{m,\ell_k^{(t)}} - \sum_{k\in[M]} c_{k,m}\mathbb{1}_{\{a_k=m\}}.$$

Note that $|\mu_m - \widehat{\mu}_m^{(t)}| \leq \epsilon_m^{(t)}$ implies that $\widehat{\mu}_m^{(t)} \leq \mu_m + \epsilon_m^{(t)}$ and $\widehat{\mu}_m^{(t)} + \epsilon_m^{(t)} \geq \mu_m$. Furthermore, $|\widehat{P}_{m,d}^{(t)} - P_{m,d}| \leq \lambda_m^{(t)}$ implies that $\widehat{P}_{m,d}^{(t)} \leq P_{m,d}+\lambda_m^{(t)}$ and $\widehat{P}_{m,d}^{(t)}+\lambda_m^{(t)} \geq P_{m,d}$. The condition $\widehat{\mu}_m^{(t)} \leq \mu_m+\epsilon_m^{(t)}$ and $\widehat{P}_{m,d}^{(t)} \leq P_{m,d}+\lambda_m^{(t)}$ yields the following upper bound of $U_m(\boldsymbol{a},\widehat{\mu}_m^{(t)}+\epsilon_m^{(t)}, \widehat{\boldsymbol{P}}_m^{(t)}+\boldsymbol{\lambda}_m^{(t)})$ as follows:

$$\begin{aligned}&U_m(\boldsymbol{a},\widehat{\mu}_m^{(t)} + \epsilon_m^{(t)}, \widehat{\boldsymbol{P}}_m^{(t)} + \boldsymbol{\lambda}_m^{(t)}) \\ &= (\widehat{\mu}_m^{(t)} + \epsilon_m^{(t)})\sum_{k\in[K]} \mathbb{1}_{\{a_k=m\}}\alpha_k(\widehat{P}_{m,\ell_k^{(t)}}^{(t)} + \lambda_m^{(t)}) - \sum_{k\in[M]} c_{k,m}\mathbb{1}_{\{a_k=m\}} \\ &\leq (\mu_m + 2\epsilon_m^{(t)})\sum_{k\in[K]} \mathbb{1}_{\{a_k=m\}}\alpha_k(\widehat{P}_{m,\ell_k^{(t)}}^{(t)} + \lambda_m^{(t)}) - \sum_{k\in[M]} c_{k,m}\mathbb{1}_{\{a_k=m\}}\end{aligned}$$

$$\leq (\mu_m + 2\epsilon_m^{(t)}) \sum\nolimits_{k \in [K]} \mathbb{1}_{\{a_k=m\}} \alpha_k (P_{m,\ell_k^{(t)}} + 2\lambda_m^{(t)}) - \sum_{k \in [M]} c_{k,m} \mathbb{1}_{\{a_k=m\}}$$

$$= U_m(\boldsymbol{a}; \mu_m, \boldsymbol{P}_m) + \mu_m \sum\nolimits_{k \in [K]} \mathbb{1}_{\{a_k=m\}} \alpha_k 2\lambda_m^{(t)} + 2\epsilon_m^{(t)} \sum\nolimits_{k \in [K]} \mathbb{1}_{\{a_k=m\}} \alpha_k (P_{m,\ell_k^{(t)}} + 2\lambda_m^{(t)})$$

$$\leq U_m(\boldsymbol{a}; \mu_m, \boldsymbol{P}_m) + 2\alpha_1 \mu_{\max} \lambda_m^{(t)} \sum\nolimits_{k \in [K]} \mathbb{1}_{\{a_k=m\}} + 2\epsilon_m^{(t)} (\alpha_1 + 2\alpha_1 \lambda_m^{(t)}) \sum\nolimits_{k \in [K]} \mathbb{1}_{\{a_k=m\}}$$

$$= U_m(\boldsymbol{a}; \mu_m, \boldsymbol{P}_m) + 2\alpha_1 (\mu_{\max} + 1)(\lambda_m^{(t)} + \epsilon_m^{(t)}) \sum\nolimits_{k \in [K]} \mathbb{1}_{\{a_k=m\}} + 4\alpha_1 \epsilon_m^{(t)} \lambda_m^{(t)} \sum\nolimits_{k \in [K]} \mathbb{1}_{\{a_k=m\}}$$

$$\leq U_m(\boldsymbol{a}; \mu_m, \boldsymbol{P}_m) + 2\alpha_1 (\mu_{\max} + 1)(\lambda_m^{(t)} + \epsilon_m^{(t)}) \sum\nolimits_{k \in [K]} \mathbb{1}_{\{a_k=m\}} + 4\alpha_1 (\mu_{\max} + 1)\epsilon_m^{(t)} \lambda_m^{(t)} \sum\nolimits_{k \in [K]} \mathbb{1}_{\{a_k=m\}}$$

$$\leq U_m(\boldsymbol{a}; \mu_m, \boldsymbol{P}_m) + 4\alpha_1 (\mu_{\max} + 1)(\lambda_m^{(t)} + \epsilon_m^{(t)} + \epsilon_m^{(t)} \lambda_m^{(t)}) \sum\nolimits_{k \in [K]} \mathbb{1}_{\{a_k=m\}}$$

$$\leq U_m(\boldsymbol{a}; \mu_m, \boldsymbol{P}_m) + 4\alpha_1 (\mu_{\max} + 1)(\lambda_m^{(t)} + 2\epsilon_m^{(t)}) \sum\nolimits_{k \in [K]} \mathbb{1}_{\{a_k=m\}},$$

where the last step follows $\lambda_m^{(t)} \leq 1$. Then it follows that

$$\Phi_m^{(t)} = U_m(\boldsymbol{a}, \widehat{\mu}_m^{(t)} + \epsilon_m^{(t)}, \widehat{\boldsymbol{P}}_m^{(t)} + \boldsymbol{\lambda}_m^{(t)}) - U_m(\boldsymbol{a}; \mu_m, \boldsymbol{P}_m) \leq 4\alpha_1 (\mu_{\max} + 1)(\lambda_m^{(t)} + 2\epsilon_m^{(t)}) \sum\nolimits_{k \in [K]} \mathbb{1}_{\{a_k=m\}}.$$

The condition $\widehat{\mu}_m^{(t)} + \epsilon_m^{(t)} \geq \mu_m$ and $\widehat{P}_{m,d}^{(t)} + \lambda_m^{(t)} \geq P_{m,d}$ yields that

$$U_m(\boldsymbol{a}, \widehat{\mu}_m^{(t)} + \epsilon_m^{(t)}, \widehat{\boldsymbol{P}}_m^{(t)} + \boldsymbol{\lambda}_m^{(t)}) \geq (\boldsymbol{a}, \mu_m, \widehat{\boldsymbol{P}}_m^{(t)} + \boldsymbol{\lambda}_m^{(t)}) \geq U_m(\boldsymbol{a}; \mu_m, \boldsymbol{P}_m).$$

Where each step is a consequence of the piece-wise monotone property of the utility function. Summing them up with respect to $m$, step I concludes.

**Step II:** Regret due to pulling suboptimal arms, but the confidence band of parameters hold. Recall that the confidence interval of parameters holding implies that

$$U(\boldsymbol{a}; \boldsymbol{\mu}, \boldsymbol{P}) \leq \text{UCB}^{(t)}(\boldsymbol{a}) \leq U(\boldsymbol{a}; \boldsymbol{\mu}, \boldsymbol{P}) + \sum_{m \in [M]} 4\alpha_1 (\mu_{\max} + 1)(\lambda_m^{(t)} + 2\epsilon_m^{(t)}) \sum\nolimits_{k \in [K]} \mathbb{1}_{\{a_k=m\}}.$$

To simplify presentation, we define:

$$\gamma^{(t)}(\boldsymbol{a}) \triangleq \sum_{m \in [M]} 4\alpha_1 (\mu_{\max} + 1)(\lambda_m^{(t)} + 2\epsilon_m^{(t)}) \sum\nolimits_{k \in [K]} \mathbb{1}_{\{a_k=m\}}.$$

Based on them, we bound the regret as:

$$\text{Reg}_T \triangleq \mathbb{E}\left[\sum_{t=1}^{T} \left(U(\boldsymbol{a}^*; \boldsymbol{\mu}, \boldsymbol{P}) - U(\boldsymbol{a}^{(t)}; \boldsymbol{\mu}, \boldsymbol{P})\right)\right]$$

$$\leq \mathbb{E}\left[\sum_{t=1}^{T} \left(\text{UCB}^{(t)}(\boldsymbol{a}^*) - U(\boldsymbol{a}^{(t)}; \boldsymbol{\mu}, \boldsymbol{P})\right)\right]$$

$$\leq \mathbb{E}\left[\sum_{t=1}^{T} \left(\text{UCB}^{(t)}(\boldsymbol{a}^{(t)}) - U(\boldsymbol{a}^{(t)}; \boldsymbol{\mu}, \boldsymbol{P})\right)\right]$$

$$\leq \mathbb{E}\left[\sum_{t=1}^{T} \gamma^{(t)}(\boldsymbol{a}^{(t)})\right]$$

$$= \mathbb{E}\left[\sum_{t=1}^{T} \sum_{m \in [M]} 4\alpha_1 (\mu_{\max} + 1)(\lambda_m^{(t)} + 2\epsilon_m^{(t)}) \sum\nolimits_{k \in [K]} \mathbb{1}_{\{a_k^{(t)}=m\}}\right]$$

$$= 4\alpha_1 (\mu_{\max} + 1)\mathbb{E}\left[\sum_{t=1}^{T} \sum_{m \in [M]} N_m^{(t)}(\lambda_m^{(t)} + 2\epsilon_m^{(t)})\right].$$

We next bound each individual term respectively. We fist bound the following term:

$$\sum_{t=1}^{T} \sum_{m \in [M]} N_m^{(t)} \lambda_m^{(t)}$$

$$= \sum_{t=1}^{T} \sum_{m \in [M]} N_m^{(t)} \sqrt{\frac{n_m^{(t)} + 1}{2} \ln \frac{\sqrt{n_m^{(t)} + 1}}{\delta}} \frac{1}{n_m^{(t)}}$$

$$\leq \sum_{t=1}^{T} \sum_{m \in [M]} N_m^{(t)} \sqrt{\frac{1}{n_m^{(t)}} \ln \frac{\sqrt{n_m^{(t)} + 1}}{\delta}}$$

$$\leq \sum_{t=1}^{T} \sum_{m \in [M]} N_m^{(t)} \sqrt{\frac{1}{n_m^{(t)}} \ln \frac{\sqrt{n_m^{(t)} + 1}}{\delta}}$$

$$\leq \sum_{m \in [M]} K \sum_{n_m^{(t)}=1}^{B_m} \sqrt{\frac{1}{n_m^{(t)}} \ln \frac{\sqrt{n_m^{(t)} + 1}}{\delta}}$$

$$\leq \sum_{m \in [M]} K \sqrt{\ln \frac{\sqrt{B_m + 1}}{\delta}} \sqrt{B_m}$$

$$\leq K \sqrt{\ln \frac{\sqrt{T}}{\delta}} \sqrt{M \sum_{m \in [M]} B_m}$$

$$= KM \sqrt{\ln \frac{\sqrt{T}}{\delta}} \sqrt{T}.$$

where $B_m$ denotes the number of rounds that arm $m$ is pulled by at least one arm when the learning ends. Similar, we obtain the following bound:

$$\sum_{t=1}^{T} \sum_{m \in [M]} N_m^{(t)} \epsilon_m^{(t)} = \sum_{t=1}^{T} \sum_{m \in [M]} N_m^{(t)} \sqrt{2\sigma^2 (\widetilde{n}_m^{(t)} + 1) \ln \frac{\sqrt{\widetilde{n}_m^{(t)} + 1}}{\delta}} \frac{1}{\widetilde{n}_m^{(t)}}$$

$$\leq \sum_{t=1}^{T} \sum_{m \in [M]} N_m^{(t)} \sqrt{4\sigma^2 \frac{1}{\widetilde{n}_m^{(t)}} \ln \frac{\sqrt{\widetilde{n}_m^{(t)} + 1}}{\delta}}$$

$$\leq 2\sigma K \sqrt{\ln \frac{\sqrt{KT}}{\delta}} \sum_{t=1}^{T} \sum_{m \in [M]} \sqrt{\frac{1}{\widetilde{n}_m^{(t)}}}$$

$$\leq 2\sigma K \sqrt{\ln \frac{\sqrt{KT}}{\delta}} \sqrt{MKT}.$$

Summing them up, we conclude the regret of this part as:

$$4\alpha_1 (\mu_{\max} + 1) KM \sqrt{T} \left( \sqrt{\ln \frac{\sqrt{T}}{\delta}} + 4\sigma \sqrt{\ln \frac{\sqrt{KT}}{\delta}} \sqrt{\frac{K}{M}} \right)$$

**Step III:** Regret due to failure of confidence bands. Note the per round regret is at most $K\mu_{\max}$. The regret is upper bounded by

$$2M(1 + d_{\max})\delta K \mu_{\max} T.$$

**Step IV:** Putting them together. Summing them up and setting $\delta = 1/T$ we have that the total regret is upper bounded by

$$\text{Reg}_T \leq 4\alpha_1(\mu_{\max} + 1)KM\sqrt{T}\left(\sqrt{\ln\frac{\sqrt{T}}{\delta}} + 4\sigma\sqrt{\ln\frac{\sqrt{KT}}{\delta}}\sqrt{\frac{K}{M}}\right) + 2M(1 + d_{\max})\delta K\mu_{\max}T$$

$$\leq 8\alpha_1(\mu_{\max} + 1)KM\sqrt{T}\left(\sqrt{\ln T} + 4\sigma\sqrt{\ln KT}\sqrt{K/M}\right) + 2M(1 + d_{\max})K\mu_{\max}$$

This proof is then complete.

### A.7 PROOF OF THEOREM 5.6

We first give a definition, which is useful in our proof.

**Definition A.1.** An action profile $\boldsymbol{a}$ is $\Psi$-optimal, if it satisfies

$$U(\boldsymbol{a}; \boldsymbol{\mu}, \boldsymbol{P}) > U(\boldsymbol{a}^*; \boldsymbol{\mu}, \boldsymbol{P}) - \Psi.$$

where $\Psi \in \mathbb{R}_+$.

**Step I:** Prove sufficient conditions on $n_m^{(t)}$, such that the selected action profile is $\Psi$-optimal.

Note that

$$U(\boldsymbol{a}^*; \boldsymbol{\mu}, \boldsymbol{P}) \leq \text{UCB}^{(t)}(\boldsymbol{a}^*) \leq \text{UCB}^{(t)}(\boldsymbol{a}^{(t)}) \leq U(\boldsymbol{a}^{(t)}; \boldsymbol{\mu}, \boldsymbol{P}) + \gamma^{(t)}(\boldsymbol{a}^{(t)})$$

This implies that

$$U(\boldsymbol{a}^{(t)}; \boldsymbol{\mu}, \boldsymbol{P}) \geq U(\boldsymbol{a}^*; \boldsymbol{\mu}, \boldsymbol{P}) - \gamma^{(t)}(\boldsymbol{a}^{(t)}).$$

Thus in each round $t$, the pulled action profile is $\gamma^{(t)}(\boldsymbol{a}^{(t)})$-optimal. The $\lambda_m^{(t)}$ can be bounded as

$$\lambda_m^{(t)} \leq \sqrt{\frac{n_m^{(t)} + 1}{2}\ln\frac{\sqrt{n_m^{(t)} + 1}}{\delta}\frac{1}{n_m^{(t)}}} \leq \sqrt{\frac{1}{n_m^{(t)}}\ln\frac{\sqrt{T}}{\delta}} \leq \sqrt{\ln\frac{\sqrt{T}}{\delta}}\sqrt{\frac{1}{n_m^{(t)}}}.$$

The $\epsilon_m^{(t)}$ can be bounded as

$$\epsilon_m^{(t)} = \sqrt{2\sigma^2(\widetilde{n}_m^{(t)} + 1)\ln\frac{\sqrt{\widetilde{n}_m^{(t)} + 1}}{\delta}\frac{1}{\widetilde{n}_m^{(t)}}} \leq 2\sigma\sqrt{\frac{1}{\widetilde{n}_m^{(t)}}\ln\frac{\sqrt{KT}}{\delta}} \leq 2\sigma\sqrt{\frac{1}{n_m^{(t)}}\ln\frac{\sqrt{KT}}{\delta}} \leq 2\sigma\sqrt{\ln\frac{\sqrt{KT}}{\delta}}\sqrt{\frac{1}{n_m^{(t)}}}.$$

Then it follows that

$$\gamma^{(t)}(\boldsymbol{a}) = \sum_{m\in[M]} 4\alpha_1(\mu_{\max} + 1)(\lambda_m^{(t)} + 2\epsilon_m^{(t)})\sum_{k\in[K]}\mathbb{1}_{\{a_k=m\}}$$

$$\leq \sum_{m\in[M]} 4\alpha_1(\mu_{\max} + 1)\sum_{k\in[K]}\mathbb{1}_{\{a_k=m\}}\left(\sqrt{\ln\frac{\sqrt{T}}{\delta}}\sqrt{\frac{1}{n_m^{(t)}}} + 2\sigma\sqrt{\ln\frac{\sqrt{KT}}{\delta}}\sqrt{\frac{1}{n_m^{(t)}}}\right)$$

$$\leq \sum_{m\in[M]} 4\alpha_1(\mu_{\max} + 1)\sum_{k\in[K]}\mathbb{1}_{\{a_k=m\}}\left(\sqrt{\ln\frac{\sqrt{T}}{\delta}} + 2\sigma\sqrt{\ln\frac{\sqrt{KT}}{\delta}}\right)\sqrt{\frac{1}{n_m^{(t)}}}$$

$$< \sum_{m\in[M]} 4\alpha_1(\mu_{\max} + 1)\sum_{k\in[K]}\mathbb{1}_{\{a_k=m\}}(2\sigma + 1)\sqrt{\ln\frac{\sqrt{KT}}{\delta}}\sqrt{\frac{1}{n_m^{(t)}}}$$

$$= \sum_{m\in[M]} 4\alpha_1(\mu_{\max} + 1)\sum_{k\in[K]}\mathbb{1}_{\{a_k=m\}}\phi(T, \delta)\sqrt{\frac{1}{n_m^{(t)}}}$$

$$= 4\alpha_1(\mu_{\max} + 1)\phi(T, \delta)\sum_{m\in[M]}\sum_{k\in[K]}\mathbb{1}_{\{a_k=m\}}\sqrt{\frac{1}{n_m^{(t)}}},$$

where $\phi(T, \delta)$ is defined as

$$\phi(T, \delta) \triangleq (2\sigma + 1)\sqrt{\ln \frac{\sqrt{KT}}{\delta}}.$$

One sufficient condition to guarantee $\gamma^{(t)}(\boldsymbol{a}) \leq \Psi$ is

$$n_m^{(t)} \geq \frac{(4\alpha_1(\mu_{\max} + 1)\phi(T, \delta)K)^2}{\Psi^2}, \forall m \in [M].$$

This can be shown by

$$\gamma^{(t)}(\boldsymbol{a}) < 4\alpha_1(\mu_{\max} + 1)\phi(T, \delta) \sum_{m \in [M]} \sum_{k \in [K]} \mathbb{1}_{\{a_k = m\}} \sqrt{\frac{1}{n_m^{(t)}}}$$

$$\leq 4\alpha_1(\mu_{\max} + 1)\phi(T, \delta) \sum_{m \in [M]} \sum_{k \in [K]} \mathbb{1}_{\{a_k = m\}} \frac{\Psi}{4\alpha_1(\mu_{\max} + 1)\phi(T, \delta)K}$$

$$= 4\alpha_1(\mu_{\max} + 1)\phi(T, \delta)K \frac{\Psi}{4\alpha_1(\mu_{\max} + 1)\phi(T, \delta)K}$$

$$= \Psi.$$

**Step 2:** We prove that when confidence bands of parameters hold, suboptimal plays make progress in identifying better action profiles. Define $\Delta$ as the utility gap between the optimal action profile and the least favored sub-optimal action profile, i.e.,

$$\Delta \triangleq U(\boldsymbol{a}^*; \boldsymbol{\mu}, \boldsymbol{P}) - \max_{\boldsymbol{a}:U(\boldsymbol{a};\boldsymbol{\mu},\boldsymbol{P}) \neq U(\boldsymbol{a}^*;\boldsymbol{\mu},\boldsymbol{P})} U(\boldsymbol{a}; \boldsymbol{\mu}, \boldsymbol{P}).$$

We divide action profiles into groups, such that the action profile in the $i$-th group $\mathcal{G}_i$ satisfies

$$U(\boldsymbol{a}^*; \boldsymbol{\mu}, \boldsymbol{P}) - 2^i\Delta < U(a) \leq U(\boldsymbol{a}^*; \boldsymbol{\mu}, \boldsymbol{P}) - 2^{i-1}\Delta.$$

Note that the the action profiles in the 0-th group are $\Delta$-optimal. In other words, all action profiles in this group are optimal action profiles. We therefore only needs to focus on the groups with index $i \geq 1$. In the following analysis, we assume $i \geq 1$. Suppose in round $t$, an action profile in $i$-group is selected, i.e., $\boldsymbol{a}^t \in \arg\max_{\boldsymbol{a} \in \mathcal{A}} \text{UCB}^{(t)}(\boldsymbol{a})$ and $\boldsymbol{a}^t \in \mathcal{G}_i$. Then it follows that there exists $m$, such that

$$n_m^{(t)} \leq \frac{(4\alpha_1(\mu_{\max} + 1)\phi(T, \delta)K)^2}{(2^{i-1}\Delta)^2}. \tag{7}$$

Let $\mathcal{M}_i \triangleq \{m | n_m^{(t)} \text{ satisfies Eq. (7)}\}$ denote a set of all arms that satisfies Eq. (7). We next prove that at least one arm from the set $\mathcal{M}_i$ is pulled, i.e., $\boldsymbol{a}^t$ makes progress in shrinking these inaccurate estimations and thus making progress for identifying better action profiles. Suppose that none of arms from $\mathcal{M}_i$ are played. We next show this leads to a contradiction. We construct an instance of the problem by that we set all the parameters of the arms to be the ground truth and the estimation error to be zero, i.e.,

$$\widehat{\mu}_m^{(t)} = \mu_m, \widehat{P}_{m,d}^{(t)} = P_{m,d}, \epsilon_m^{(t)} = 0, \lambda_m^{(t)} = 0, \forall m \in \mathcal{M}_i.$$

We left the parameters of other arms, i.e., $m \in [M] \setminus \mathcal{M}_i$ as the estimated ones. Let $\text{UCB}_i^{(t)}(\boldsymbol{a})$ denote the upper confidence of action profiles in this constructed instance. The first consequence is that this constructed instance locating a $2^{i-1}\Delta$ policy, i.e.,

$$U(\boldsymbol{a}_i^*; \boldsymbol{\mu}, \boldsymbol{P}) > U(\boldsymbol{a}^*; \boldsymbol{\mu}, \boldsymbol{P}) - 2^{i-1}\Delta,$$

where $\boldsymbol{a}_i^* \in \arg\max \text{UCB}_i^{(t)}(\boldsymbol{a})$. Note that the $\text{UCB}_i^{(t)}(\boldsymbol{a})$ is piece-wisely increasing and the upper confidence of each parameter is larger than its ground truth, which yields $\text{UCB}_i^{(t)}(\boldsymbol{a}) \leq \text{UCB}^{(t)}(\boldsymbol{a})$. Then it follows that

$$\text{UCB}^{(t)}(\boldsymbol{a}^{(t)}) \geq \text{UCB}^{(t)}(\boldsymbol{a}_i^*) \geq \text{UCB}_i^{(t)}(\boldsymbol{a}_i^*) \geq U(\boldsymbol{a}_i^*; \boldsymbol{\mu}, \boldsymbol{P}) > U(\boldsymbol{a}^*; \boldsymbol{\mu}, \boldsymbol{P}) - 2^{i-1}\Delta.$$

This contradicts that $\boldsymbol{a}^{(t)}$ belongs to the $i$-th group.

**Step III:** Regret due to pulling suboptimal arms, but the confidence bands of parameters hold. In total there are $M$ arms. By the argument of Step II, the total number of rounds that one profile from group $\mathcal{G}_i$ is selected is upper bounded by

$$M\frac{(4\alpha_1(\mu_{\max}+1)\phi(T,\delta)K)^2}{(2^{i-1}\Delta)^2}$$

rounds. Each such play incurs a regret of at most $2^i\Delta$. Thus the total regret is upper bounded by

$$\sum_{i=1}^{\infty}M\frac{(4\alpha_1(\mu_{\max}+1)\phi(T,\delta)K)^2}{(2^{i-1}\Delta)^2}2^i\Delta = \sum_{i=1}^{\infty}M\frac{(4\alpha_1(\mu_{\max}+1)\phi(T,\delta)K)^2}{(2^{i-1}\Delta)^2}2^i\Delta$$

$$= M(4\alpha_1(\mu_{\max}+1)\phi(T,\delta)K)^2\sum_{i=1}^{\infty}\frac{1}{2^{i-2}\Delta}$$

$$\leq 4M(4\alpha_1(\mu_{\max}+1)\phi(T,\delta)K)^2\frac{1}{\Delta}.$$

**Step IV:** The regret due to the failure of confidence bands. The regret is upper bounded by

$$2M(1+d_{\max})\delta K\mu_{\max}T.$$

**Step V:** The total regret. By selecting $\delta = 1/T$, we bound the total regret as:

$$\mathrm{Reg}_T \leq 4M(4\alpha_1(\mu_{\max}+1)\phi(T,1/T)K)^2\frac{1}{\Delta} + 2M(1+d_{\max})K\mu_{\max}$$

$$\leq 64MK^2\alpha_1^2(\mu_{\max}+1)^2(\phi(T,1/T)K)^2\frac{1}{\Delta} + 2M(1+d_{\max})K\mu_{\max}$$

$$\leq 64MK^2\alpha_1^2\left((2\sigma+1)\sqrt{\ln T\sqrt{KT}}\right)^2\frac{1}{\Delta} + 2M(1+d_{\max})K\mu_{\max}$$

$$= 64MK^2\alpha_1^2(2\sigma+1)^2\ln T\sqrt{KT}\frac{1}{\Delta} + 2M(1+d_{\max})K\mu_{\max}$$

$$\leq 96MK^2\alpha_1^2(2\sigma+1)^2\frac{1}{\Delta}\ln KT + 2M(1+d_{\max})K\mu_{\max}.$$

This proof is then complete.

## B  ADDITIONAL EXPERIMENTS

### B.1  MORE DETAILS ON EXPERIMENT SETTING

**Parameter setting**. The probability mass function is defined as

$$p_{m,d} = \begin{cases} \alpha d, & \text{if } d \leq \lceil m/2 \rceil, \\ \alpha(m+1-d), & \text{if } \lceil m/2 \rceil < d \leq m, \\ 0, & \text{otherwise} \end{cases}$$

where $\alpha = 1/(\sum_{d=1}^{\lceil m/2 \rceil} d + \sum_{d=\lceil m/2 \rceil+1}^{m} m+1-d)$ is the normalizing factor. The probability function exhibits a shape akin to a normal distribution. Essentially, as the index $m$ increases, there is an expected augmentation in the number of units of resource associated with an arm. This trend arises due to the shifting of probability masses towards larger values of $m$ with the increase in the index $d$. Each arm's rewards are sampled from Gaussian distributions, i.e., $\boldsymbol{R}_m \sim N(\mu_m, \sigma^2)$, where $\mu_m \in [1,2]$ and $\sigma > 0$. We examine three cases regarding the reward mean:

- **Inc-Shape**: $\mu_m = 1 + m/M$, the reward mean increases with the index of arm $m$.
- **Dec-Shape**: $\mu_m = 2 - m/M$, the reward mean decreases with the index of arm $m$.
- **U-Shape**: $\mu_m = 1 + |M/2 - m|/M$, the reward mean initially decreases and then increases with the index of arm $m$.

We designate the movement cost as $c_{k,m} = \eta |(k \mod M) - m| / \max\{K, M\}$, where $\eta \in \mathbb{R}_+$ is a hyper-parameter that controls the scale of the cost. Unless explicitly varied, we adopt the following default parameters: $T = 10^4$, $\delta = 1/T$, $K = 10$ plays, $M = 5$ arms, $\eta = 1$, $\sigma = 0.2$ and the U-Shape reward. Furthermore, the number of play types is 2, with parameter $\alpha = [3, 1]$.

### B.2 ADDITIONAL EXPERIMENTS

**Impact of resource-reward correlation** we fix the probability mass function of resource and three cases of $\mu_m$, i.e., Inc-Shape (positive correlation), Dec-Shape (negative correlation), and U-Shape (weak correlation). Fig. 3 shows the corresponding regret of our algorithms and the baselines. In Fig. 3a, it is evident that the regret curves for `MSB-PRS-ApUCB` under Inc-Shape, U-Shape, and Dec-Shape initially exhibit a sharp increase before plateauing, indicating a sub-linear regret. Additionally, when $\mu_m$ follows Inc-shape pattern, the convergence rate of regret is slower compared to cases where it follows Dec-shape or U-shaped pattern. Fig. 3b illustrates that the regret curves for `OnlinActPrf` and `OnlinActPrf-v` follow a linear trend, while the regret curve for `MSB-PRS-ApUCB` consistently remains at the bottom. This observation confirms that `MSB-PRS-ApUCB` yields the smallest regret compared to the two baseline algorithms. This trend persists even when $\mu_m$ is under U-Shape and Dec-Shape, as shown in Fig. 3c and 3d, respectively.

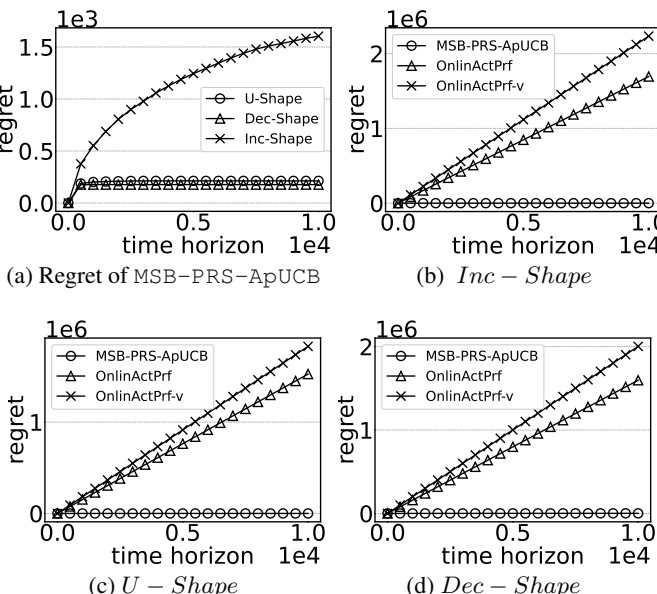

Figure 3: Impact of Resource-reward Correlation.

**Impact of movement cost** We varied the number of plays, denoted as $\eta$, across three settings: $\eta = 1$, 2, and 10, and plotted the regret of three algorithms. In Fig. 4a, it is evident that the regret curves for `MSB-PRS-ApUCB` under $\eta = 1$, 2, and 10 initially exhibit a sharp increase before plateauing, indicating a sub-linear regret. Additionally, Fig. 4b illustrates that the regret curves for `OnlinActPrf` and `OnlinActPrf-v` follow a linear trend, while the regret curve for `MSB-PRS-ApUCB` consistently remains at the bottom. This observation confirms that `MSB-PRS-ApUCB` yields the smallest regret compared to the two baseline algorithms. This trend persists even when $\eta = 2$ and 10, as shown in Fig. 4c and 4d, respectively.

**Impact of the standard deviation of reward** We varied the standard deviation of reward, denoted as $\sigma$, across three settings: $\sigma = 0.1$, 0.2, and 0.3, and plotted the regret of three algorithms. In Fig. 5a, it is evident that the regret curves for `MSB-PRS-ApUCB` under $\sigma = 0.1$, 0.2, and 0.3 initially exhibit a sharp increase before plateauing, indicating a sub-linear regret. Additionally, Fig. 5b illustrates that the regret curves for `OnlinActPrf` and `OnlinActPrf-v` follow a linear trend, while the regret curve for `MSB-PRS-ApUCB` consistently remains at the bottom. This observation confirms

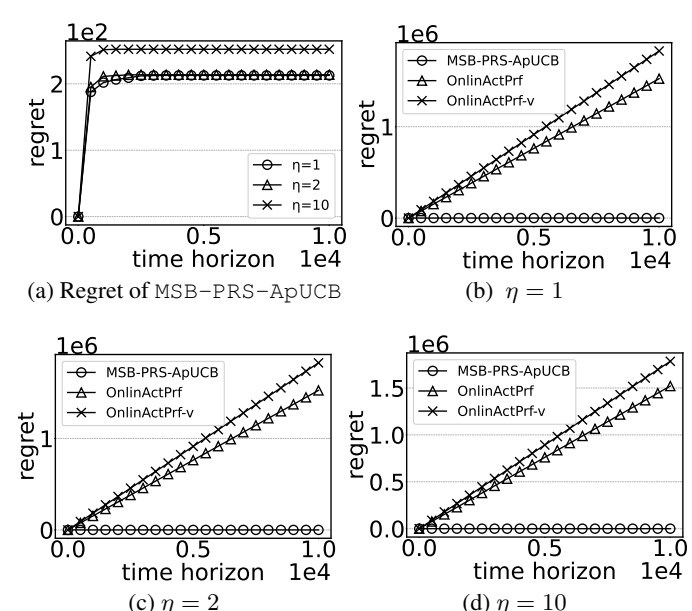

Figure 4: Impact of Movement Cost.

that `MSB-PRS-ApUCB` yields the smallest regret compared to the two baseline algorithms. This trend persists even when $\sigma = 0.2$ and $0.3$, as shown in Fig. 5c and 5d, respectively.

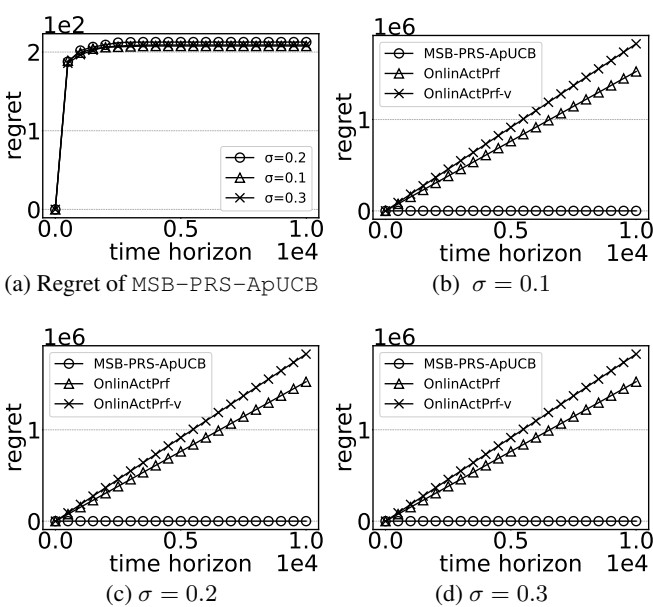

Figure 5: Impact of Standard Deviation of Reward.

