# OpenReview forum: "Multiple-play Stochastic Bandits with Prioritized Resource Sharing"
_ICLR.cc/2025/Conference — Submitted to ICLR 2025_

### Official Review · Reviewer_Rvss · 2024-11-01

**Soundness:** 3
**Presentation:** 2
**Contribution:** 2
**Rating:** 5
**Confidence:** 3

**Summary:**

This paper considers a variant of the stochastic bandit problem where players can select multiple arms from a pool consisting of a fixed number of arms, with each arm's capacity following a time-invariant distribution. Players receive rewards only if the chosen arms have sufficient corresponding capacity. The objective is to maximize cumulative rewards over a fixed-horizon game. To address this problem, the paper proposes a new algorithm based on the philosophy of combinatorial bandits, along with learning the capacity distribution, under the assumption of an oracle's existence. For the proposed algorithm, the paper provides both lower and upper bound analyses on the regret to demonstrate its (near) optimality. Numerical experiments are conducted to validate the proposed algorithm and demonstrate improvements compared to benchmarks.

**Strengths:**

1. The paper considers a novel problem setting where the arm capacity is stochastic, unlike existing work.
2. The paper also develops algorithms specifically to address this proposed problem setting.
3. The theoretical effectiveness of the proposed algorithm is supported through both lower and upper bounds.
4. The numerical experiments help illustrate the algorithm’s performance.

**Weaknesses:**

1. The paper mentions a use case for this problem setting in the LLM context. However, I am curious if this could be more practical, specifically whether it is something that could feasibly be deployed in that context.

2. The existence of an oracle depends on locating the maximum weight matching, which is referenced from existing work. I wonder if this reference includes any theoretical guarantee supporting the claim that this oracle is theoretically optimal. Further justification would be beneficial here.

3. The lower bound analysis lacks technical novelty.

4. The real challenge posed by stochastic capacity is unclear. Existing work assumes deterministic arm capacity without requiring it to be known. It is difficult to assess whether the stochastic realization actually makes the problem more challenging (due to randomness) or possibly easier (given known observations).

**Questions:**

I would refer to the weakness part. Any responses/comments would be very helpful.

---

### Official Review · Reviewer_oKpD · 2024-11-02

**Soundness:** 1
**Presentation:** 3
**Contribution:** 2
**Rating:** 5
**Confidence:** 4

**Summary:**

The paper extends the multi-play multi-armed bandit (MP-MAB) model to include a prioritized resource-sharing mechanism, referred to as MSB-PRS. The model targets resource allocation scenarios in LLM and edge intelligence applications, where different plays are assigned different priorities, and each arm has multiple but random capacities. The authors establish both instance-independent and instance-dependent regret lower bounds for the model and propose an efficient learning algorithm, MSB-PRS-ApUCB, which achieves order-optimal regret bounds. The authors also conduct simulations based on synthetic data to validate their proposed algorithm.

**Strengths:**

1. The introduction of prioritized resource sharing into the multi-play bandit framework is novel, nabling random arm capacities and differentiated priorities for various plays, which are well-suited for practical applications such as LLM and edge intelligence.

2. The proposed MSB-PRS-ApUCB algorithm is thoughtfully designed and well-motivated, achieving regret bounds that closely align with the established lower bounds, up to acceptable factors.

3. The synthetic experiments provide a good assessment of the performance of MSB-PRS-ApUCB compared to baseline algorithms.

4. The paper is well-structured and clearly written, making it easy to follow.

**Weaknesses:**

1.  The learning component of the algorithm and the regret analysis are fairly standard, as there exists an optimal matching between players and arms, and the remaining thing is to target this optimal matching through UCB strategy, as done in much of the literature. However, I acknowledge that finding the optimal matching is not easy due to the nonlinear combinatorial structure of the utility functions.

2.  The concentration bound in Lemma 5.4 seems incorrect. I believe the authors should use Lemma 9 from [Maillard, et al., 2017] instead of Lemma 10. Consequently, the inequalities in lines 977, 1006, and 1053 also appear to be incorrect. The authors should check these carefully.

**Questions:**

1. In line 190, the reward is defined as being scaled by the priority parameter. Could the authors provide a practical example to clarify this definition?

---

### Official Review · Reviewer_cHu6 · 2024-11-03

**Soundness:** 2
**Presentation:** 2
**Contribution:** 3
**Rating:** 3
**Confidence:** 3

**Summary:**

This paper presents a new framework called Multiple-Play Stochastic Bandits with Prioritized Resource Sharing (MSB-PRS), which belongs to the research area of multi-play multi-armed bandit. Within this framework, an efficient algorithm is developed to identify the optimal play allocation policy while maintaining low computational complexity. The study establishes lower bounds for both instance-independent and instance-dependent regret. Additionally, the proposed algorithm is based on the application of the classic Upper Confidence Bound (UCB). It maintains the same per-round computational complexity and achieves sublinear regret upper bounds that closely align with the established lower bounds.

**Strengths:**

The multi-play multi-armed bandit (MP-MAB) problem is a significant model in online learning, and I appreciate the efforts that the authors intest in solving an interesing model of it, that is the MSB-PRS problem. For it, an algorithm has been developed to identify the optimal play allocation policy with a specific complexity. The upper bounds on regret are close to the lower bounds (up to some factors) in both instance-dependent and instance-independent scenarios.

**Weaknesses:**

My main concern is that while this work provides rigorous theoretical analysis and proofs, I am still not entirely clear on its contributions.

First, although the problem model is somewhat introduced, I find it challenging to connect it with specific examples of resource allocation. While the authors mention its applicability in high-interest areas like LLMs, they do not provide corresponding explanations. Is there a way to contextualize its application in LLMs, or could examples of practical applications be included?

Second, the authors offer some related work, but I still struggle to compare them with this study. To address this, I suggest including a table to compare the results of this work with previous findings.

Finally, the experiments are overly simplistic; they do not thoroughly describe the experimental setups or compare it with other studies. While I appreciate the authors' efforts in deriving theoretical results, I believe there is still significant room for improvement in the presentation of this work.

**Questions:**

I do not have any questions.

---

### Official Review · Reviewer_CXEK · 2024-11-04

**Soundness:** 3
**Presentation:** 2
**Contribution:** 2
**Rating:** 5
**Confidence:** 3

**Summary:**

The paper works within the MP-MAB framework and proposes a variant of the framework catered to LLM and edge intelligence applications. With these applications in mind, the work imposes additional structure in the form of their MSB-PRS Model.

In Section 3 they introduce the model and provide the problem formulation. In Section 4 they characterize the hardness of the problem and fundamental learning limits by providing lower bounds, In Section 5 they present their UCB based learning algorithms, and in Section 6 they present experiments validating their approach and comparing it to baselines from the literature.

**Strengths:**

1. New variant of MP-MAB with resource prioritization.
2. Lower bounds characterizing the hardness of their problem variant
3. UCB based algorithm for learning - ApUCB
4. Instance dependent and instance independent upper bounds on ApUCB

**Weaknesses:**

In my opinion while there seem to be innovative and impactful ideas in the paper it can use a lot better presentation before being accepted. In this bullet I would highlight how Section 3 on the MSB-PRS Model is barely readable by being cluttered by endless notation. Such a presentation makes it incredibly hard to takeaway any intuitive mental pictures of the setup that could then serve as the basis of appreciating the methods presented in the remaining paper.

Actionable Suggestions:
Please add a high-level overview paragraph in Section 3 before introducing the model mathematically using the complete notation. Please include an illustrative example or visual representation of the MSB-PRS model alongside this new paragraph.

2. The paper does a poor job of motivating their target applications with the exposition being limited to a few lines in the introduction with vague wordings. In particular the only reference to LLM applications reads the following in the introduction: "in LLM applications, reasoning tasks and LLM instances can be modeled as plays and arms respectively. .... priority quantified by price, membership hierarchy". Which by itself gives very little insight into how the modeling in the paper is good for this application.

I would encourage the authors to expand on their motivating example by dedicating a sub section to explaining how the MSB-PRS model applies to LLM and edge intelligence applications.

**Questions:**

1. On pg 8 around Eqn 6 the authors try to say that computing exact-UCB is intractable and introduce UCB instead. In the process they say that "Exact-UCB may attain the max value at different selections of \mu, P for different action values especially when the confidence band fails". It is not clear at all by what is meant by this sentence and better explanation and writing are needed in this regard.

2. What are the baselines OnlinActPrf and -v version doing exactly? If this paper is the one introducing this new MSB-PRS framework structure then how was this approach from the literature adapted to make a fair comparison with your novel approach? These details are currently missing from both Section 6 and Appendix B.

I would suggest that the authors provide a brief description of the OnlinActPrf and OnlinActPrf-v algorithms in Section 6 or Appendix B explaining how they were adapted to the MSB-PRS setting. Additionally, please discuss the fairness of the comparison given the differences in problem formulation.

---

### Meta-Review · Area_Chair_5jsU · 2024-12-08

**Metareview:**

This paper presents a new framework called Multiple-Play Stochastic Bandits with Prioritized Resource Sharing (MSB-PRS), which belongs to the research area of multi-play multi-armed bandit. Within this framework, an efficient algorithm is developed to identify the optimal play allocation policy while maintaining low computational complexity. There are many concerns raised by the reviewers for motivation and contribution, which are not addressed by the authors.

**Additional Comments On Reviewer Discussion:**

NA

---

### Decision · Program_Chairs · 2025-01-22

Reject